# A Comparative Study of Software Defined Networking Controllers Using Mininet

**Neelam Gupta** [1], **Mashael S. Maashi** [2], **Sarvesh Tanwar** [1], **Sumit Badotra** [3], **Mohammed Aljebreen** [4] **and Salil Bharany** [5,*]

1   Amity Institute of Information Technology, Amity University Uttar Pradesh, Noida 201301, India
2   Software Engineering Department, College of Computer and Information Sciences, King Saud University, Riyadh 11451, Saudi Arabia
3   Department of Computer Science and Engineering, Lovely Professional University, Phagwara 144001, India
4   Department of Computer Science, Community College, King Saud University, P.O. Box 28095, Riyadh 11437, Saudi Arabia
5   Department of Computer Engineering & Technology, Guru Nanak Dev University, Amritsar 143005, India
*   Correspondence: salil.bharany@gmail.com

**Abstract:** Software Defined Networking (SDN) is a relatively new networking architecture that has become the most widely discussed networking technology in recent years and the latest development in the field of developing digital networks, which aims to break down the traditional connection in the middle of the control surface and the infrastructure surface. The goal of this separation is to make resources more manageable, secure, and controllable. As a result, many controllers such as Beacon, Floodlight, Ryu, OpenDayLight (ODL), Open Network Operating System (ONOS), NOX, as well as Pox, have been developed. The selection of the finest-fit controller has evolved into an application-specific tool operation due to the large range of SDN applications and controllers. This paper discusses SDN, a new paradigm of networking in which the architecture transitions from a completely distributed form to a more centralized form and evaluates and contrasts the effects of various SDN controllers on SDN. This report examines some SDN controllers or the network's "brains," shows how they differ from one another, and compares them to see which is best overall. The presentation of SDN controllers such as Ryu, ODL, and others is compared by utilizing the Mininet simulation environment. In this study, we offer a variety of controllers before introducing the tools used in the paper: Mininet. Then, we run an experiment to show how to use ODL to establish a custom network topology on a Mininet. The experimental results show that the O controller, with its larger bandwidth and reduced latency, outperforms other controllers in all topologies (both the default topology and a custom topology with ODL).

**Keywords:** software defined networking; controller; infrastructure plane; mininet; application layer; control plane

## 1. Introduction

Traditional networks are constrained by a range of service requirements as well as the network's size [1]. These requirements relate to engineered traffic, management of flow, policy implementation, reliability, and some of these include virtualization. Networks have become increasingly complicated and difficult to design and operate. With enabler technologies like cloud computing, portability, and emergent conception like the IoT (Internet of Things), the current tendency is to connect everything. Apart from extra bandwidth, these innovative ideas demand a network that is simpler and more resilient, and this in turn requires a better data centre infrastructure. Networking, a new grid system, has been designed to address traditional network difficulties in a novel way. Web server downtime, network speed quality, data mislay, and other classic network concerns are all common. For a web executive, traditional network troubleshooting is time-consuming

and difficult. SDN [2–9] is a new networking paradigm that arose in response to the drawbacks that traditional networks have demonstrated. It is used to make networks more programmable and to make network management easier. It allows traditional networks to separate their control and data plane functions, resulting in a more dynamic, adaptable, automated, and manageable design. SDN's concepts, namely active nets and control and infrastructure surface uncoupling, are not newly discovered; they are the outcome of past exploration and Stanford University's [3] invention of the OpenFlow protocol. Despite the fact that the core proposal was not innovative, it has the benefit that the setting was suitable. Control logic proceeds to the SDN controller, which runs an operating system (OS) on top of what is known as a Data Plane Network (DNTN), where devices are connected to each other by wires and cables. The network is programmable by software applications that interface with data plane devices, which become basic forwarding elements and do not have direct physical control over them [10–15]. It disassociates control functionality from the network plane tool and centralizes it, having effective traffic forwarding and flow direction over the estate, in contrast to traditional networks. Figure 1 illustrates a multi-dimensional architecture with data forwarding devices on the end user plane and controllers configuring them on the monitor surface. The control layer interacts with the prominent application/management plane to programme the entire network and enforce various regulations. Interfaces allow distinct levels to communicate with one another via communication/programming protocols. The Infrastructure Surface is the part of the web where end-user data are transported. The three layers of SDN networks are the infrastructure layer (Data Plane), the control layer (Control Plane), and the application layer (Application Plane) (Management Plane) [15–19]. This means it has complete control over the network system. Control, data, and application are the three logical planes that it divides the network into. The control plane, sometimes known as the network's brain, is a central management unit that acts as a controller. SDN applications can manage practically any type of network traffic [20–27]. As a result, it faces numerous challenges, such as SDN layer-by-layer security from the forwarding layer to the application plane [28–34].

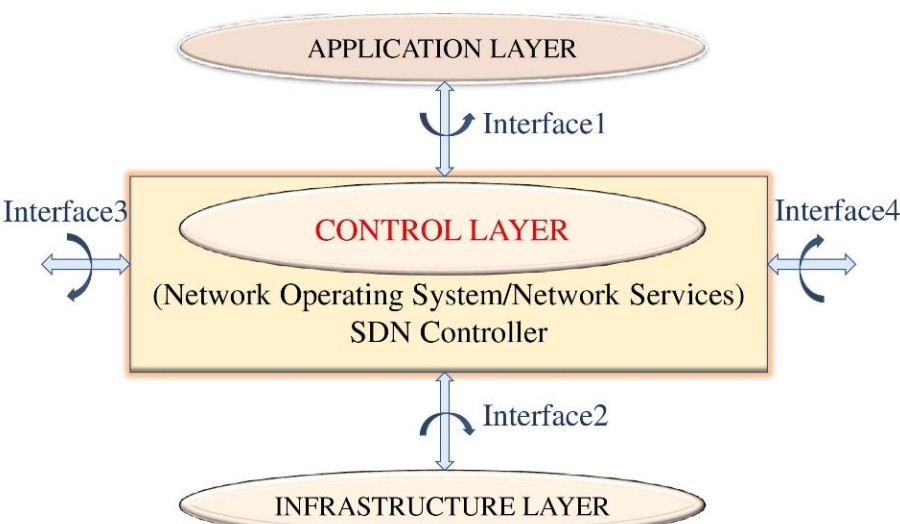

**Figure 1.** Architecture of SDN layer.

In SDN architecture, an SDN controller is an app that is in charge of flow control. A web controller is a device that allows operators to choose between different types of networks, as well as the QoS and privacy of domains [35–41]. It can be found in data centres, wireless networks, broad-area and fundamental grids, as well as smart homes and other connected devices [4–6]. There are two types of SDN controllers: those for data centre NFV (Network Function Virtualization) and those for managing network switches. SDN controllers can perform a variety of roles in the SDN architecture, including allowing

servers to instruct switches on where packets should be delivered. Pox, ODL, ONOS, Ryu, Trema, Floodlight, and NOX [7–9] are all popular SDN controllers that are written in distinct languages (Pox uses Python, Trema uses Ruby, and ODL uses Java., for example), as shown in Ta4 and have varying performance and applications. The data plane is made up of the modems and bridging hubs that make up the client/server architecture. They are now simply forwarding devices [42–48]. The control plane is the part of a net that connects the data and control planes and provides features such as firewall and load balancing that were previously provided by middleboxes. In a Software Defined Wide Area Network (SD-WAN) network, the control plane [10] acts as a hub for all data and traffic to and from the network. An open southbound interface is employed for communication between the northbound and southbound parts of the system. For this interface, the OpenFlow protocol [10,11] is now the most frequently acknowledged option. Other interfaces can be utilized, such as Open vSwitch Database (OVSDB), Network Configuration (NETCONF) Protocol, Simple Network Management (SNMP) convention, and so on. There is yet to be an industry-wide agreement on a northbound coupling allying the SDN controller and apps. SDN controllers are classified according to the programming language used to construct them, the time it takes a programmer to understand how to create applications for that controller, the types of southbound interfaces it provides, and so on [49–54]. On the mininet testbed, the proposed research will identify how different controllers function and establish a default (single, linear, tree, reverse, and simple) and one custom topology. We chose one of the most widely used SDN controllers today as a test case (default or reference controller ODL). Because of the significance of the controller in the SDN architectonics, as well as the variety of planning and performances available on the market, it is necessary to assess and compare all of these options against various presentations. The demonstration of SDN controllers such as Ryu, ODL, and others is compared by utilizing the Mininet simulation environment. The controller makes the majority of the forwarding decisions before moving down to the switches and executing logical judgments. The data move amid the Application surface and the Data surface is managed by a network controller using Southbound Application Program Interfaces (APIs) and Northbound APIs. Advantages include worldwide control and observation of the entire network [12] at once, useful automation of operations such as network operation, better server and network utilization, and so on. This study's major contributions are summarized as follows: The architecture compares and contrasts the effects of various SDN controllers on SDN as it moves from a fully distributed to a more centralized form [55–62]. Using the mininet simulation environment, we conducted an experiment to create a default topology as well as a custom topology. In our search, we employed terms such as keywords and index/subject terms. The investigation is based on works located in the computerized Scopus database. The search is limited to the years 2012 through 2022. The graph summarizes the research on SDN controller comparison that has been done during the past nine years, including information about the authors, the country, the sources, etc.

The following paper is assembled as follows: Section 2 presents related work and background and workflow of Software Defined Networking layers. An SDN controller timeline over the last nine years is presented in Section 3. Section 4 presents a comparison of various controllers with various (default and a custom topology) experiments on the Mininet emulator, and in Section 5, the SDN controller features are described. Section 6 includes some concluding observations.

## 2. Related Work and Background

The security of the Internet and its applications remains an unsolved research problem. The difficulties have long been recognized by the interface research association and corporation. Previously, Networking of Named Data, networks with programmability, Hypertext Transfer Protocol as the narrow midsection, and SDN were all mentioned as fresh concepts for enhancing the design of future networks. It is being hailed as the most

auspicious solution for the Internet of the future. SDN is defined in numerous ways, each with its own set of characteristics and ramifications for the ICT industry [63–66].

SDN [4] is a ground-breaking network architecture that decouples the network control and forwarding, allowing it to be directly programmable. Rather than complex networking systems, SDN provides simple programmable network devices. Network control may be separated from data flows in this architecture, and data can be programmable on both planes at the same time—a critical component of SDN [13]. SDN architecture has the following characteristics, as shown in Table 1. Customers who have experienced quick changes in network demand may benefit from this as they look for new ways to manage their data centre's capacity [67–73].

**Table 1.** SDN architecture [5] Features.

| Sr_No. | Features | Description |
| --- | --- | --- |
| 1 | Programmable Directly | Network control may be directly designed since it is separated from forwarding services. |
| 2 | Managed from The Centre | Network intelligence is (logically) centralized in software-based SDN controllers that maintain a unified image of the network. |
| 3 | Acrobatic | Administrators can manage network-wide traffic flow flexibly to suit changing demands by abstracting control from forwarding. |
| 4 | Configured Programmatically | SDN enables network administrators to easily manage, configure, protect, and optimize network resources. |
| 5 | Vendor Neutral | SDN controllers, not vendors or protocols, provide SDN deployment instructions. |

The severance of the control and forwarding layers makes the grid more programmable and gives external software more freedom to govern how the network behaves. This feature's combination could enable better configuration, performance, and support for network architectural and operational innovation. The differences between Software Defined Networking and Traditional Networking are shown in Figure 2. SDN [14] allows users to easily and flexibly adopt new ideas, applications, and revenue-generating services thanks to its high configurability. It is possible to clearly separate virtual networks, allowing for real-world testing. To introduce novel concepts in stages, a straightforward transition from the investigational to the operational phase might be adopted.

*Workflow of SDN Layers*

Data Plane: Switching devices make up the data plane's infrastructure layer (e.g., switches, routers, etc.). These devices capture network data, store it on local devices for a short time, and then send it to controllers. Second, they oversee packet processing in line with controller-defined rules. The net rank may comprise facts like network structure, statistics on traffic, and network applications [15,16]. The physical surface is the part of the SDN [17] network that comprises all devices, sensors, and gateways that are connected to the network via cables, mobile phones, and other devices. The physical layer, shown in Figure 3, includes all SDN networking devices that are responsible for traffic forwardings, such as Hubs, Switches, Routers, and Bridges.

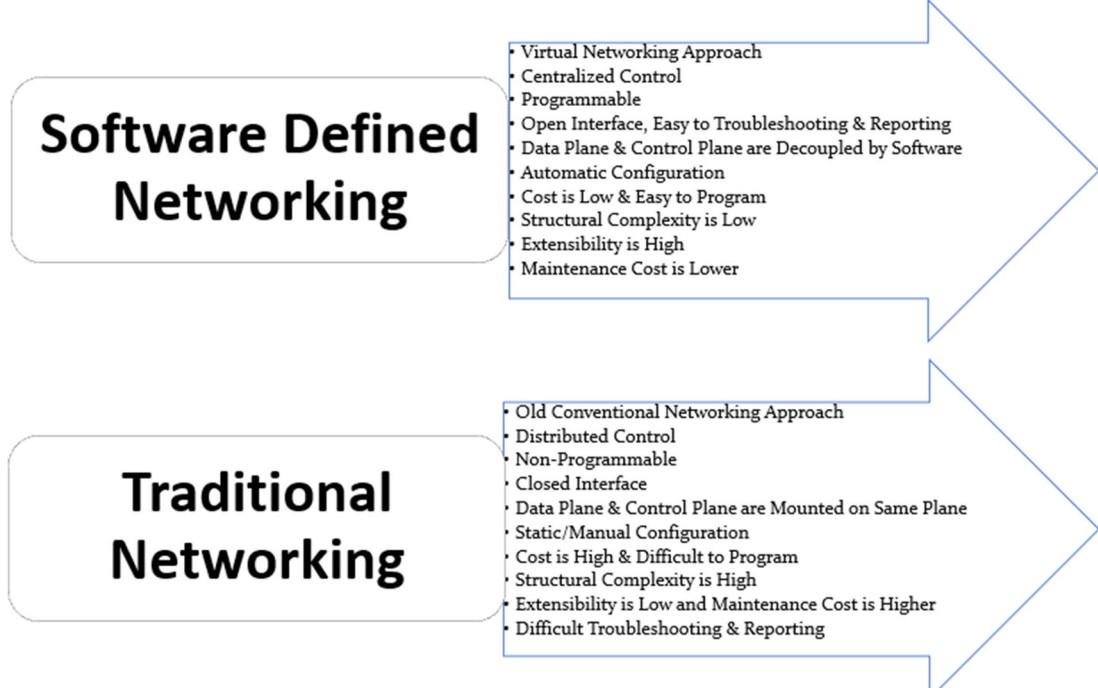

**Figure 2.** Differences between Software Defined Networking and Traditional Networking.

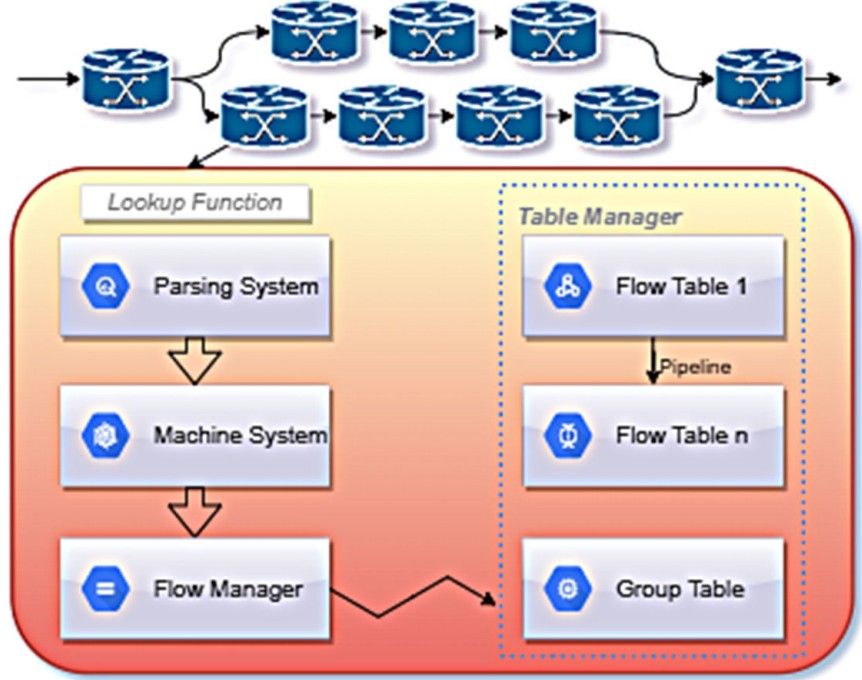

**Figure 3.** Workflow of data plane.

Control Plane: The control layer defines functions that controllers can use to access the switching device functionality. Two examples of potential outcomes involve network stability reports and packet forwarding rules import. SDN programmes can use an interface, such as an application programme interface, to get system-level data from the switching appliance. Communication interfaces will be needed in the time ahead for a big administrative net domain to be administered by a single central controller, according to the US Department of Defense (DoD) and the European Union (EU) ETSI Network

Operations Centre. They can also utilize this information to make decisions about system optimization [18]. It sits between the physical and application layers, managing devices and providing physical layer developer APIs. It also provides aggregated data, network transfer, and analysis for urban sensing. Figure 4 shows the control plane, which includes SDN controllers and interacts with applications to transmit the data flow to its destination. At this layer, the SDN controller ensures device sharing and QoS-aware data routing, and it is where devices are connected to each other and interact with applications.

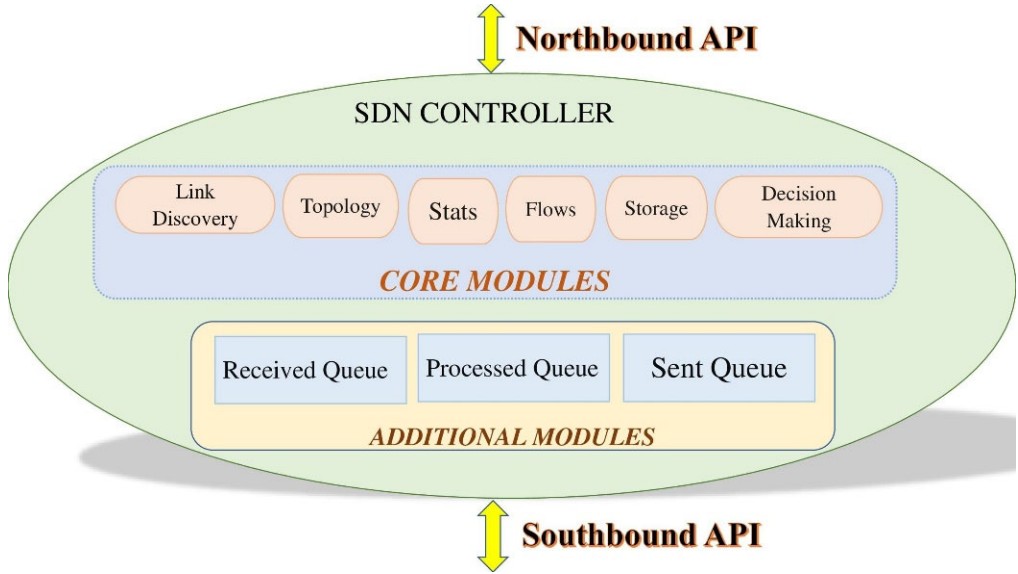

**Figure 4.** Workflow of control plane.

Application Plane: At the infrastructure layer, [19,20] the switching appliance can be accessed and managed by SDN applications. It allows developers to create a range of high-performance Internet of Things (IoT) applications that link directly to the web. SDN apps include the control of dynamic access, continuous movement, mobility, server load distribution, and virtualization of networks. As a result, the physical infrastructure—such as cables, power lines, and other components—is largely hidden from view. The application layer provides network apps that users can use to submit data regarding network-specific requests on demand, as depicted in Figure 5.

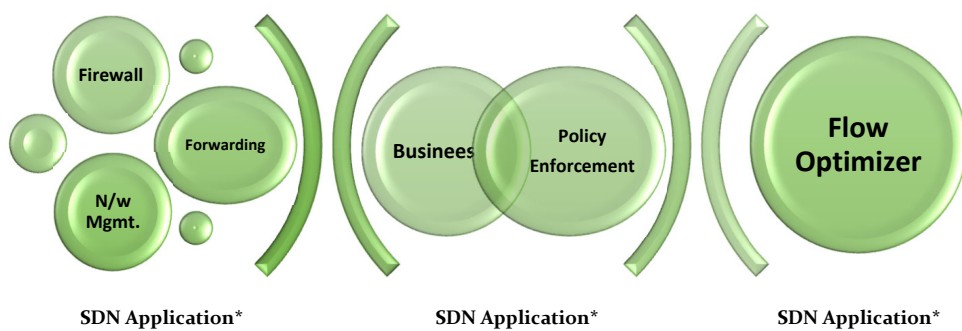

**Figure 5.** Workflow of application plane.

In Figure 5. The workflow of application plane is provided, this workflow is including varioys SDN applications and flow optimizer. SDN is the latest development in the field of developing software that can supply a range of end-to-end conclusions for data centre and enterprise nets. The types of apps that can be constructed include automation of networks, network configuration, and management, [20] network surveillance, troubleshooting a network, network regulations, and stability, to name a few. Brocade and HPE (Hewlett Packard

Enterprise), for example, have the following highly useful applications, as illustrated in Figure 6.

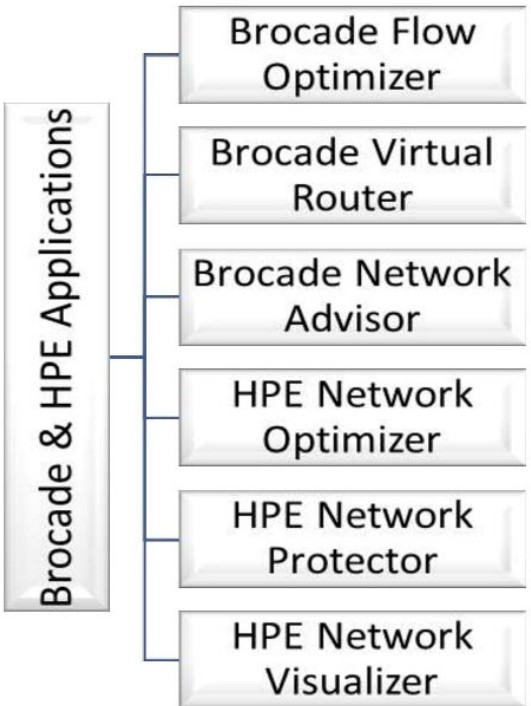

**Figure 6.** Brocade and HPE applications.

Southbound Interface: The control links allying the controller and infrastructure plane tools such as physical and virtual switches and routers are defined by southbound protocols. This is due to the protocol's dependency on the hardware on which it operates. It enables dynamic communication between high-level programmes and the system's actual/virtual tools.

Northbound Interface: The controller is connected to control programmes (also known as applications) via the northbound interface (or web OS). Applications can utilise these APIs to manage (or programme) the data layer to do activities, namely engineering for traffic, the discovery of topology, balancing the workload, packet filtering, latency and delay mitigation, and more.

East and West Bound Interface: Within an SDN, the east and west coupling governs how controllers connect with one another in order to share data. Data can also be sent between servers or between servers and controllers via this interface. SDN is significant since this transmission route is critical for large companies [21–25].

## 3. SDN Controller Timeline

The SDN Controller is the most important component for making traffic management decisions in underlying networks. Routing, switches, firewall, translating network addresses, and load-balancing are just a few of the functions it may perform with the forwarding elements [21] (data plane switches). Topology and traffic flow are the controller's primary functions. Packet-out messages are used by the link discovery module to send inquiries to external ports. The topology manager is responsible for keeping the topology in good shape. To discover the optimum paths between network nodes, decision-making is used. A multi-controller architecture, as depicted in Figure 7, is a group of controllers working together to accomplish a certain standard of achievement and manageability.

**A multi controller architecture**

- Basics modules: link discovery module, topology module, storage module, strategy making module, flow table module and control data

- Two modules are responsible for providing the routing service: the topology manager, the link discovery modules

- Link Discovery module is responsible for discovering & maintaining the status of physical links in the netwrok. The information collected is used to build the neighbor database in the controller, capturing all the OpenFlow neighbors

- Link discovery between OpenFlow nodes using traditional link layer discovery protocol LLDP & link discovery between edge OpenFlow node and Host. This procedure is triggered by the controller when any unknown traffic enters the OpenFlow domain

- The topology manager builds and maintains the topology information in the controller & calculate the routes inthe network. This modules uses the neighbor database to compute the network topologies based on the recieved information from the link discovery module

- The topology manager builds the global topology database at the controller, which contains the shortest path information to any OpenFlow node or host

**Figure 7.** A multi-controller architecture.

Networking is a modern networking solution with a centralized control plane. The term 'distributed control plane' refers to how all networking devices' control planes are contained within the device. The SDN controller is a device that monitors the activities of all networking devices at the same time. The distributed control plane is used by all traditional networking equipment [19,20] such as routers and switches. This is because it is impossible to have a single control plane for all networking equipment. It links resources to management applications and performs device-to-device flow operations based on application policies. Although each device has its own control plane, they are all managed by the same device, as shown in Figure 8. A controller [3] can operate as a hub for controlling these resources since it can see the entire network, including data plane devices. SDN is a novel method of providing data to the control plane, where all calculations are done, and a variety of applications and capabilities can be attached as required. Because of the controller's relevance in SDN design, as well as the range of architectures and implementations on the market, it is vital to analyze and compare all of these solutions against numerous performance indicators. In SDN [22], the data and control surfaces are divided, permitting them to communicate wirelessly and over the internet.

The attempt to centralize network control is not new, as seen by modern SDN controllers and SDN architecture. Multiple attempts have been made to segregate the control logic from the data plane since the mid-2000s. Both a vast variety of matching header fields and a wide variety of functionality are absent from these previous designs' control planes. OpenFlow is an open protocol that encourages software developers to create applications on various switches that support flow tables with a wide variety of header data. By enabling server virtualization, quick reaction to network changes, the deployment of policies, and centralized control over the entire network, SDN promotes flexibility and agility.

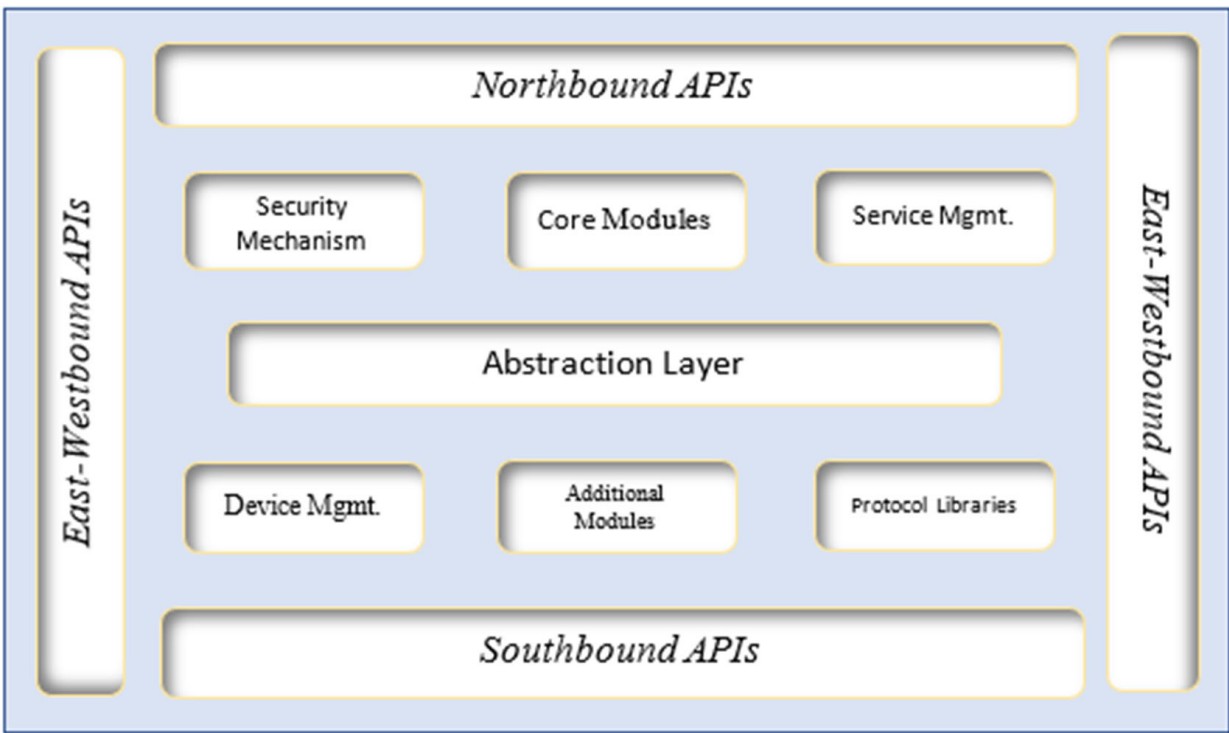

**Figure 8.** SDN controller architecture.

SDN controllers are the 'brains' of a software-defined network [23–28]. They control the flow of data connecting the switches and routers, 'below', and the apps and business logic, 'top'. It may perform a variety of simple duties, as well as more complicated ones like running analytic algorithms and coordinating new rules across the network. Searching is an important component of performing a systematic review since it gives you a foundation for your study. The search approach should be thorough and objective, as well as simple and repeatable. The work done on the comparison of SDN controllers in the last 9 years is given below, shown in Figure 9a–f. In our search, we employed terms such as keywords (authors, keywords, and countries) and index/subject (relevant sources, clusters) terms. This investigation is based on publications found in the Scopus electronic database. In this study, the quality of library services has been examined in relation to publishing and citation patterns during the previous 11 decades (2012–2022). The top authors, nations, organizations, journals, types of collaboration, highly cited publications, etc., are also examined in this bibliometric study. Data extraction from the Scopus databases has been performed using it. A thorough search strategy is developed to extract the most pertinent data sources. Excel, Biblioshiny, Cite Space, and the VOS viewer software have all been used for data analysis. It has been discovered that 353 items in total are published in Scopus throughout this time. India is reported to be the most productive country, with the most productive organizations, authors, and author partnerships. This appears to be the first thorough bibliometric analysis that integrated productivity and citations, centrality, and citation impact to offer an overall picture of the literature's contribution to library service quality. Most studies were published in the year 2019 (*n* = 54), and this year had the highest amount of research published. The search is restricted from 2012 to 2022, as shown in Figure 10a–c.

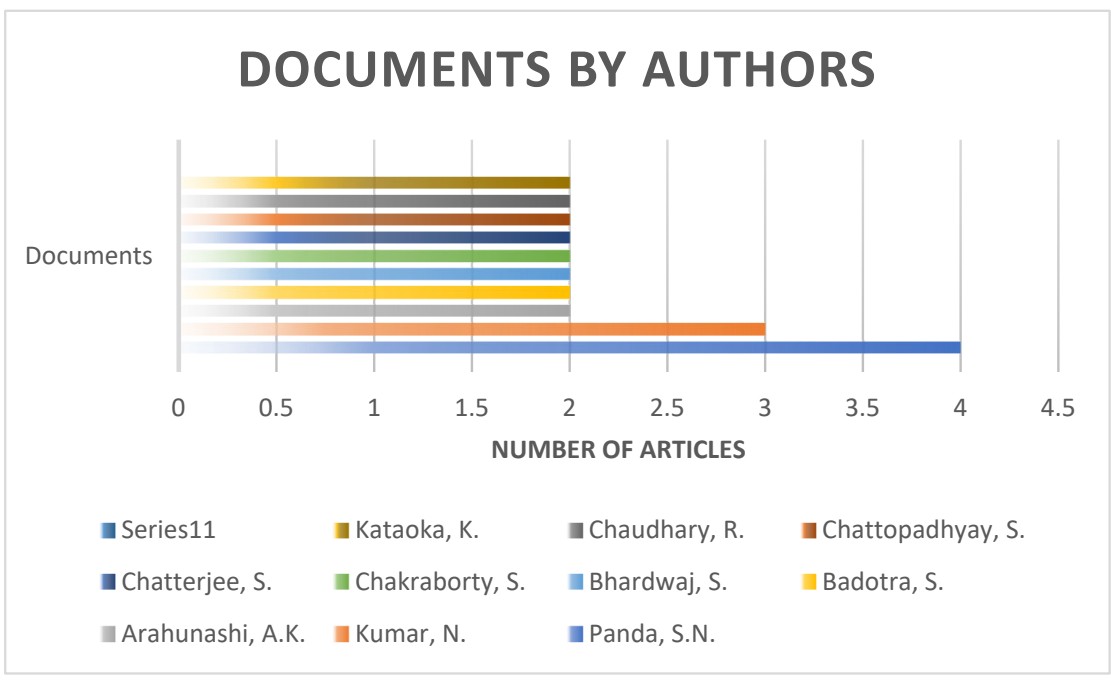

(**a**)

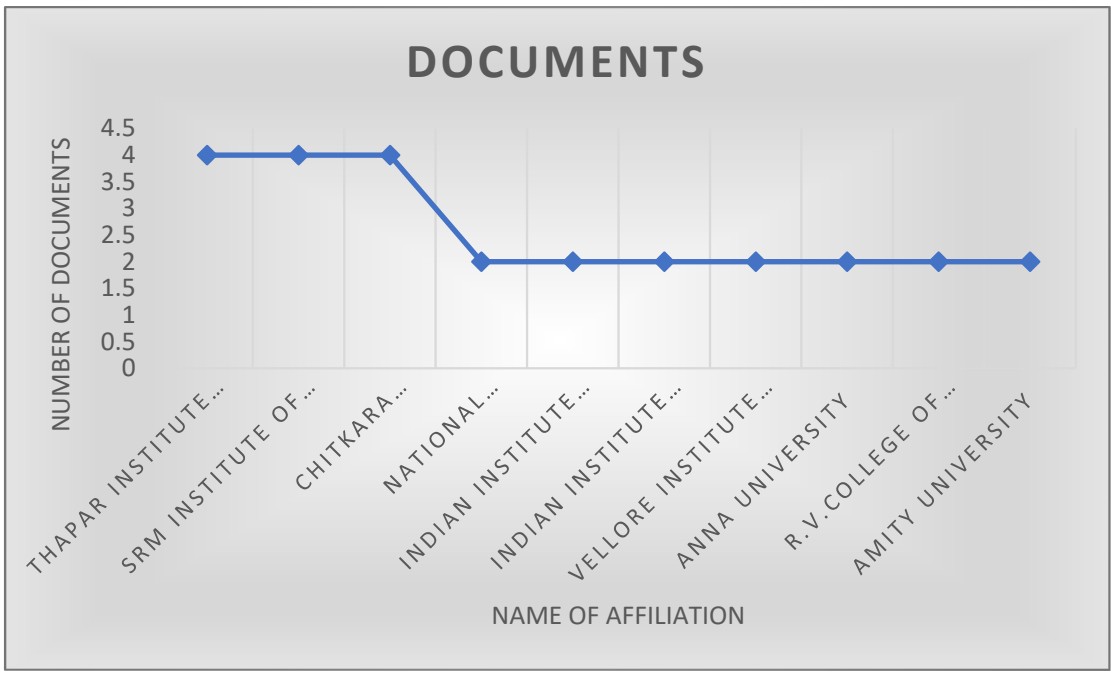

(**b**)

**Figure 9.** *Cont.*

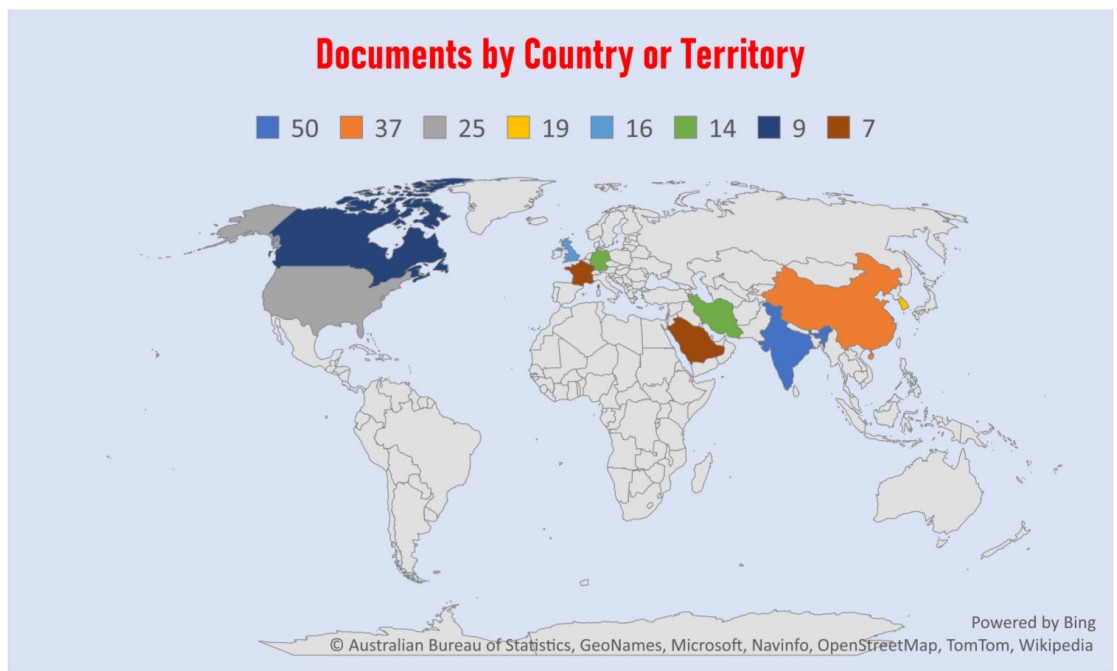

(**c**)

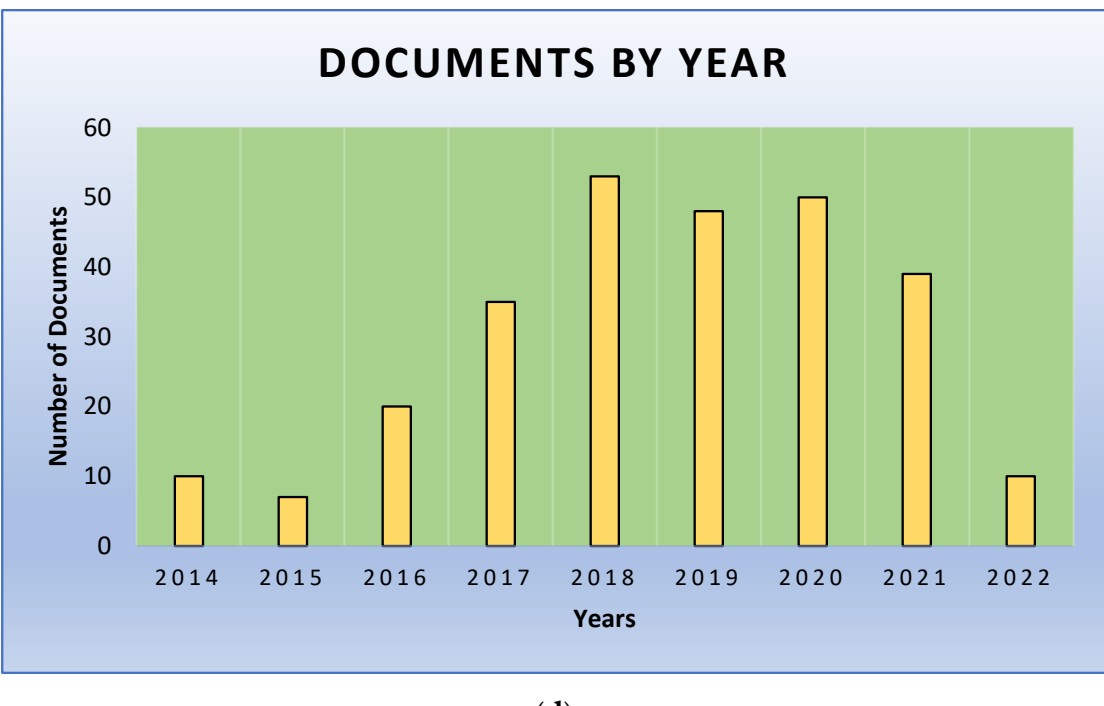

(**d**)

**Figure 9.** *Cont.*

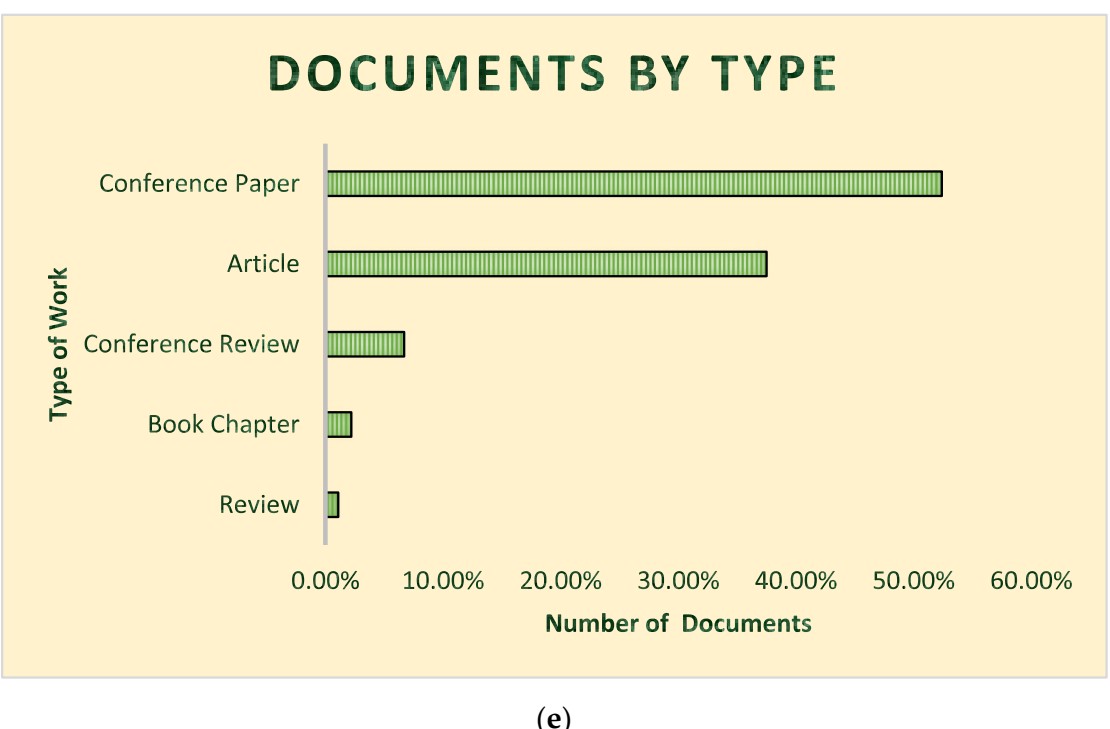

(**e**)

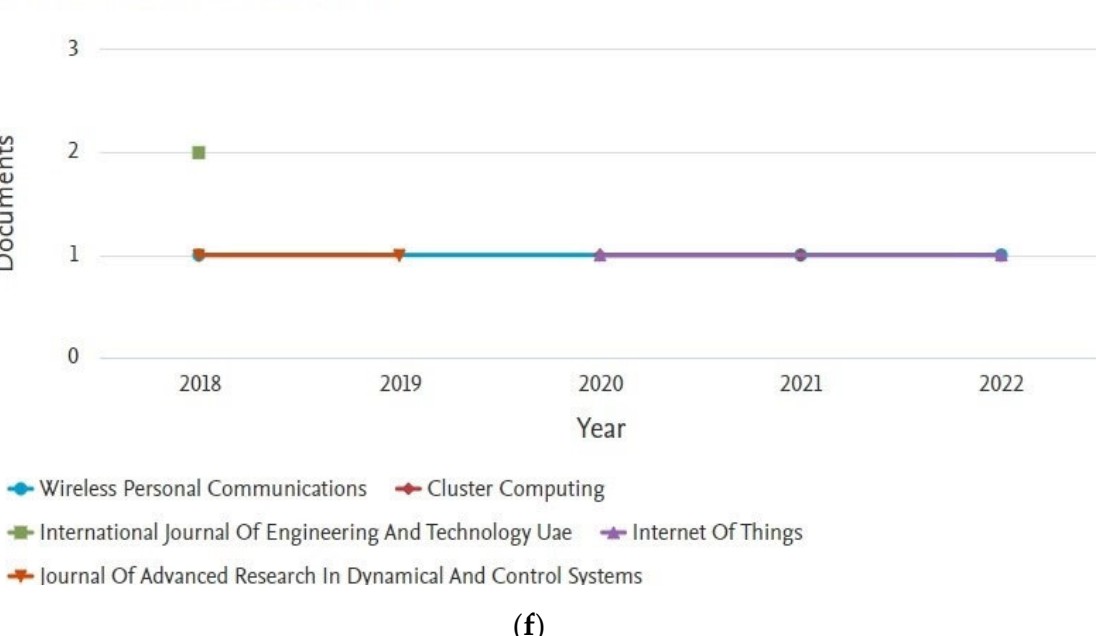

(**f**)

**Figure 9.** (**a**): The authors and cumulative numbers of research articles related to SDN controllers. (**b**): The affiliation and cumulative numbers of research documents related to SDN controllers. (**c**): The country and cumulative numbers of research documents related to SDN controllers. (**d**): The annual and cumulative numbers of research documents related to comparison SDN controllers. (**e**): The type of work and cumulative numbers of research documents related to SDN controllers. (**f**): The annual with source and cumulative numbers of research documents related to SDN controllers.

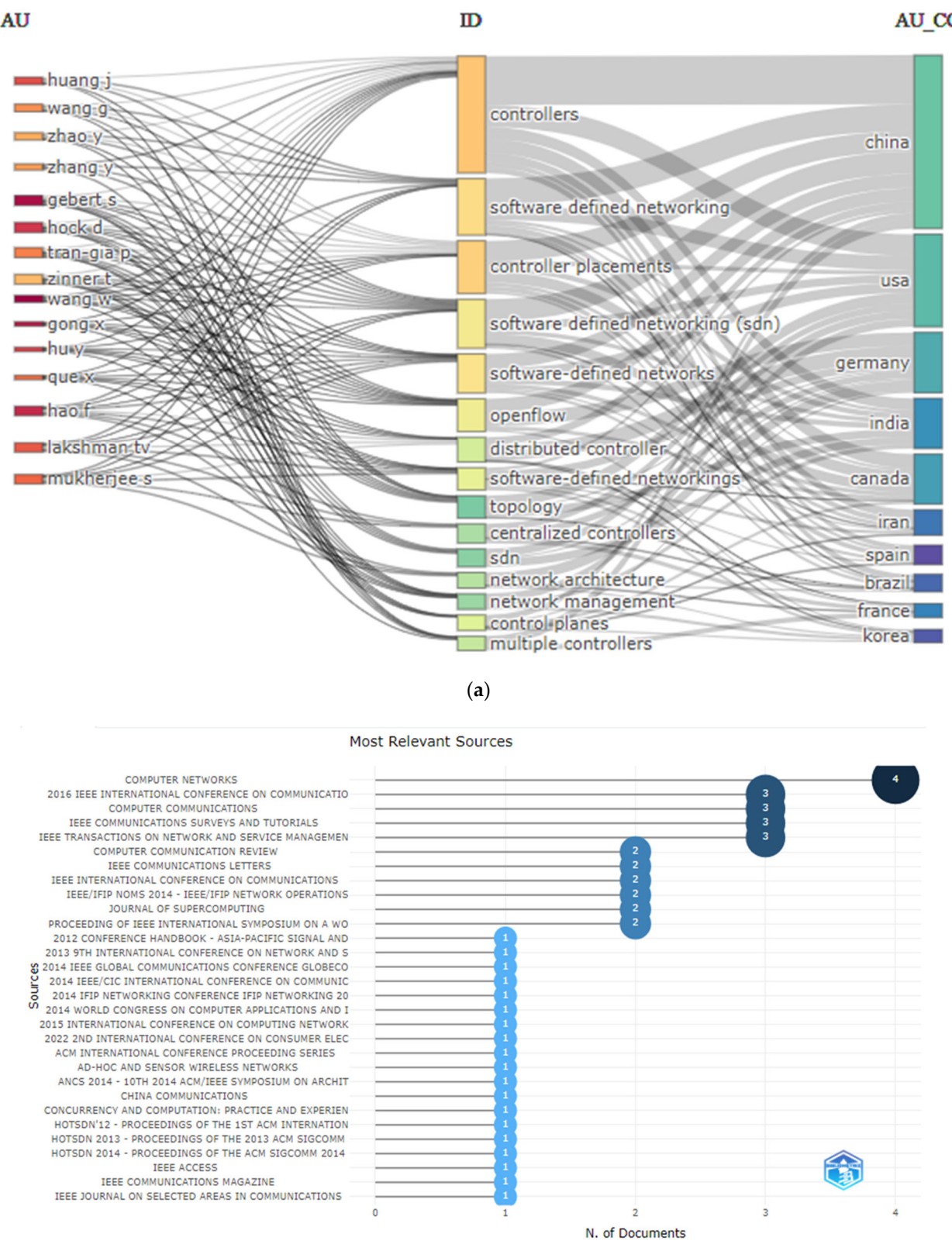

(**a**)

(**b**)

**Figure 10.** *Cont.*

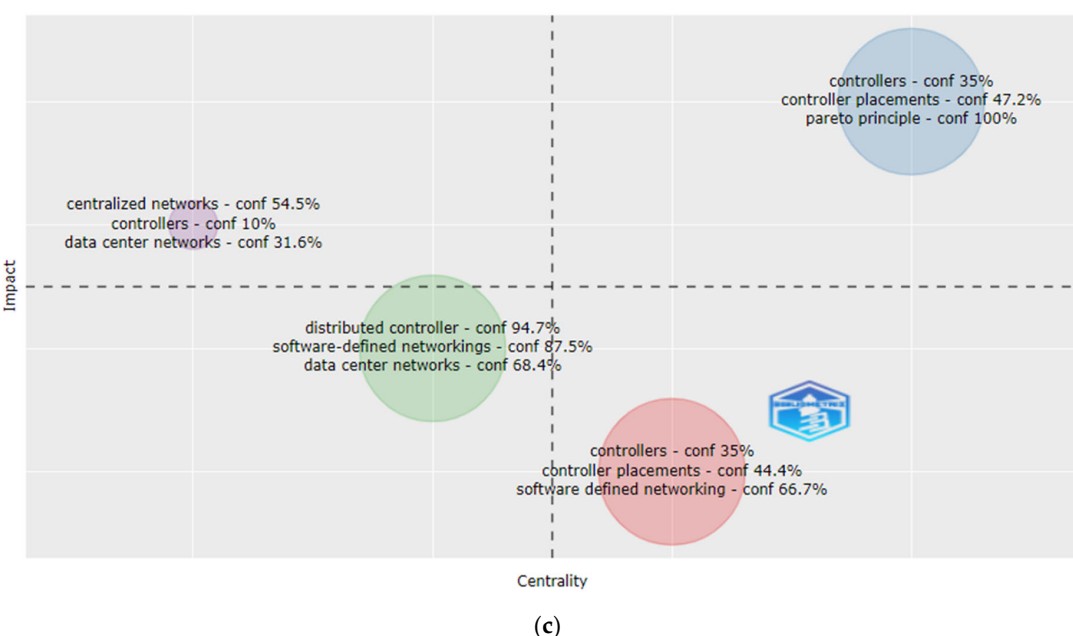

(**c**)

**Figure 10.** (**a**): Three fields-plot between the authors, keywords plus, and countries through Biblioshiny. (**b**): Most relevant sources and cumulative numbers of research documents related to SDN controllers. (**c**): Clusters by authors coupling.

## 4. Comparison of Controllers

As illustrated in Figure 11, there are several Open Source SDN controllers on the market, each with its own unique use case. However, how they are used is based on a number of norms, as well as the capability matrix [28–38] and the cultural fit of the organization and project [38–50]. In this post, we will look at the maturity of some of the most prominent Open-Sourced controllers in industry and academia, as well as some of the most well-known designs using the controller's API and Platform, as shown in Tables 2 and 3.

**Table 2.** Controller platform.

| Reference | Controller Name | License | Architecture | Programming Language | Documentation | Support Platform |
|---|---|---|---|---|---|---|
| [33] | ONOS | Apache 2.0 | Distributed Flat | Java | Good | Linux, macOS, Windows |
| [34] | Trema | GPL 2.0 | Centralized | C, Ruby | Fair | Linux |
| [35] | Ryu | Apache 2.0 | Centralized | Python | Good | Linux, macOS |
| [36] | RUNOS | Apache 2.0 | Distributed Flat | C++ | Fair | Linux |
| [37] | FloodLight | Apache 2.0 | Centralized | Java | Good | Linux, macOS, Windows |
| [38] | Maestro | LGPL 2.1 | Centralized | Java | Limited | Linux, macOS, Windows |
| [39] | Mul | GPL 2.0 | Centralized | C | Good | Linux |
| [40] | ODL | EPL 1.0 | Distributed Flat | Java | Good | Linux, macOS, Windows |
| [42] | POX | Apache 2.0 | Centralized | Python | Limited | Linux, macOS, Windows |
| [38] | NOX | GPL 2.0 | Centralized | C++ | Limited | Linux |
| [34] | Beacon | GPL 2.0 | Centralized | Java | Fair | Linux, macOS, Windows |
| [35] | Faucet | Apache 2.0 | Centralized | Python | Good | Linux |

**Table 3.** Controller's API.

| SDN Controller | POX | NOX | OpenDayLight | Beacon | RUNOS | FloodLight | Faucet | Maestro | OpenMul | Trema | RYU | ONOS |
|---|---|---|---|---|---|---|---|---|---|---|---|---|
| NORTHBOUND API | ad-hoc | ad-hoc | REST, RESTCONF, XMPP, NETCONF | ad-hoc | REST | REST, JavaRPC, Quantum | - | ad-hoc | REST | ad-hoc | REST | REST, Neutron |
| SOUTHBOUND API | OpenFlow 1.0 | OpenFlow 1.0 | OpenFlow 1.0, 1.3 | OpenFlow 1.0 | OpenFlow 1.3 | OpenFlow 1.0, 1.3 | OpenFlow 1.3 | OpenFlow 1.0 | OpenFlow 1.0, 1.3, OVSDB, Netconf | OpenFlow 1.0 | OpenFlow 1.0–1.5 | OpenFlow 1.0, 1.3 |
| EAST/ WESTBOUND API | - | - | Akka, Raft | - | Maple | - | - | - | - | - | - | Raft |
| MULTITHREADING | No | Yes | Yes | Yes | Yes | Yes | Yes | Yes | Yes | - | Yes | Yes |
| CONSISTENCY | No | No | Yes | No | Yes | Yes | Yes | No | No | No | Yes | Yes |
| MODULARITY | Low | Low | High | Fair | High | Fair | - | Fair | High | Good | Fair | High |

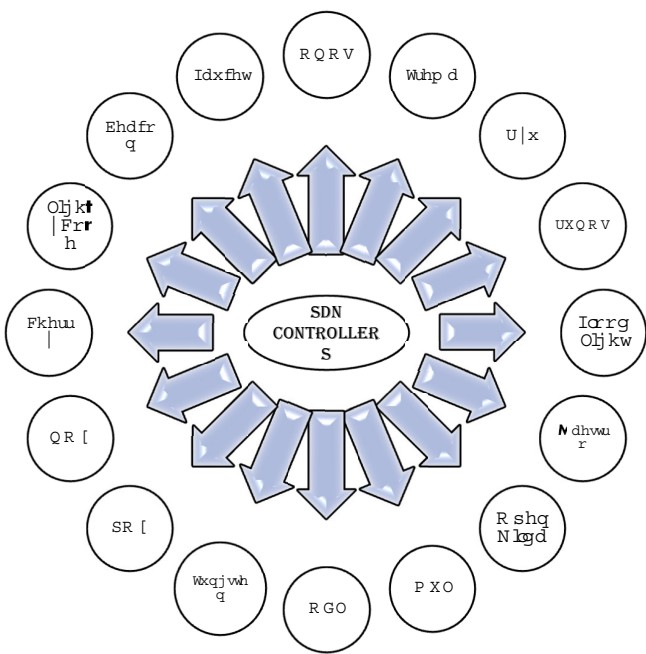

**Figure 11.** SDN controller.

### A.    *Beacon*

Beacon [25] is a Java-based OpenFlow control plane that assists OpenFlow 1.0, and it was founded in 2010 as a Stanford University research project. It is small, cross-platform, and modular, and it supports threading and activity processes. It is a beacon-based device used extensively in education and research. Beacon was created with three key goals in mind: speed, application development simplicity, and runtime configuration. It was originally designed with Linux in mind but has been upgraded to work with Windows and Mac. It uses the Open Services Gateway Initiative (OSGi) protocol and Equinox to deliver often sought aircraft capabilities. Beacon makes use of the OSGi, Spring Framework, and Spring Web frameworks. Multithreaded programming is used to boost performance, particularly while reading and writing messages to switches. Beacon [26] is a server that can run for months without shutting down and has the processing power to power a data centre. It is modular, platform-independent, and extremely quick. Without interfering with other packages, code packages can be installed even while they are operating. Beacon might also enable custom UI frameworks and integrated webserver businesses.

### B.    *Faucet*

Faucet [27] is an open-source OpenFlow controller that lets network managers manage their networks the same way they manage server clusters. Faucet replaces traditional routers or switches embedded firmware with vendor-independent server-based software. It allows network managers to move network control functions (such as routing protocols, neighbour finding, and switching algorithms). Faucet is an SDN controller that enables anyone to build completely customizable networks. Faucet [28] is a small, open-source OpenFlow controller that allows users to administer their networks in the same way as server clusters are managed. The OpenFlow project has been open-source since 2004 when a series of conferences were hosted in Auckland, New Zealand. Reannez designed the first version of Faucet using prototype code provided by the University of Waikato's WAND Network Research Group. Faucet is a browser-based tool that allows you to manage, test, and extend your network architecture. It is managed by a YAML configuration file that provides the network topology and network functions that are necessary. The appropriate OpenFlow programming is then performed on each network device. Because many vendors use Faucet as part of their QA process, cross-vendor interoperability is now a viable option.

### C. *Lighty core*

Lighty core is an OpenDaylight-powered Software Development Kit that facilitates the development of SDN solutions in Java easier and faster [29]. You can use whatever framework you want, not just Karaf because the Lighty core provides Java SE as a runtime. Memory management and speed have been improved, as well as the ability for your application code to start, stop, and restart modules as needed.

### D. *Cherry*

The Software-Defined Networking (SDN) [30] project is designed for IT service providers employing SDN, not for general use. Among the qualities of the cherry are the following: It supports the OpenFlow 1.0 and 1.3 protocols [31–35], with a focus on interoperability with commercially available OpenFlow switches. It has a simple plugin system for northbound programmes and a RESTful API for controlling the controller.

### E. *NOX*

NOX [36–42] is a network control platform that provides administration solutions and newer control applications with a flexible high-level interface. NOX is part of the software-defined networking (SDN) ecosystem, which is a pre-defined framework for constructing OpenFlow-based network control apps. OpenFlow, the first SDN technology to be given a name, was created by Nicira Networks. Its purpose is to provide a platform for developers and academics to create innovative apps for industrial and business networks. It was developed at the University of California, Berkeley, and comes in two flavours: original and multi-threaded. It is written in C++. NOX [32] has been the basis for a multitude of research initiatives in the early phases of the SDN sector since Nicira initially made it accessible to researchers in 2008.

### F. *POX*

Open source and Python-based, POX [40–45] is a control application platform for software-defined networking (SDN). The POX offers a framework for connecting to SDN switches using OpenFlow or OVSDB. The Python programming language may be used to develop an SDN controller. Use the Command Line Interface to invoke POX [41] directly. This system employs a RADIUS authentication server with IEEE 802.1X, with the Auth Flow POX controller routing all requests from the virtual routers to the authentication server. POX is likely to divulge information since it can identify incorrect ARP header field values. In NBI, authentication [43] may prevent a DoS attack, but in SBI, rate restriction and packet dropping, among other safeguards, can lessen the danger. A DoS attack can be triggered by sending a large number of bogus packets to the controller. Spoofing is exceedingly improbable after using the Auth Flow authentication method.

### G. *ONOS*

The ONOS [43–47] controller for telecommunications networks is a Java-based open-source controller. The target performance has been specified to be exceedingly excessive, like 500 k-1 M track frame-ups per second or 500 M-1 T operations per second. The ONOS distributed core surface handles controller synchronization through replication. A command line interface (CLI), online GUI views, and a REST-API are all options for interacting with the ONOS controller. Core model objects are protocol agnostic and are used to describe networks and states. ONOS depicts networks using directed graphs. The Northbound and Southbound APIs allow users to control their devices. ONOS [35], an open-source distributed network OS, is another SDN controller. Its NBI may be configured in two ways: prescriptive or fine-grained, or intent-based, where users simply define their requirements. ONOS is compatible with Open Flow versions 1.0 and 1.3, as well as other agreements such as OVS DB and OF config.

### H.    Tungsten Fabric

Tungsten Fabric (previously OpenContrail) is an open-source SDN [36,37] controller that provides connectivity and security for virtual, containerized, and bare-metal applications. The majority of TF's features are platform and device agnostic, and it can connect VM-container-legacy stacks. To connect it to your existing infrastructure, Professional Services (PS) may be necessary. TF Config is the most popular TF component, having the most developers working on it. In a nutshell, it is a database that stores all setups. The network administrator simply needs to indicate the intended behaviour of the network, not all of the variables. The remaining bits are put together automatically. You could want to enable network communication [38–41] between two networks, for example. The creation of networks is automated using an intent-based strategy. When you use TF, everything is easy because the majority of the settings are default and completed instantly [42,43]. A firewall is unblocked, allowing traffic between the two networks to flow freely, and devices receive all of the necessary network configuration information.

### I.    OpenDaylight

OpenDaylight [48–50] is an open-source software initiative that intends to improve SDN by introducing industry-standard protocols and functions. It is free to join and acts as a collaborative platform for anybody interested in SDN, including end users and consumers. The OpenFlow protocol creates an open communications interface for controllers to communicate with the data progressing surface, and it was the first SDN standard. OpenFlow and other SDN protocols are supported by ODL [46], a free software initiative. This protocol gives businesses more control over their networks and allows them to adapt to changing needs. A modular controller framework is supported, as well as numerous SDN standards and protocols. Apps can collect data from the network, perform analytics algorithms, and distribute new policies across the network. The Beacon controller inspired the ODL project, which uses OSGi to enable more plug-ins. ODL is a distributed controller that offers high availability in cluster topologies (HA). Other protocols supported by ODL include NETCONF, RESTCONF, Yang, and BGP. In June 2015, security was added to the Lithium release. The evaluation rates of the products according to the following Table 4, which is based on our weighted criteria-based scoring: ODL is 84.5%, and Ryu is 73.2%.

**Table 4.** Evaluation criteria between ODL and Ryu.

| Criterion | Weight | ODL | Ryu |
| --- | --- | --- | --- |
| Typical Architecture | 3.0 | 2.4 | 2.4 |
| Operations Support | 5.0 | 2.5 | 2.5 |
| OpenFlow Support | 20.0 | 19.0 | 20.0 |
| API Support | 30.0 | 30.0 | 24.0 |
| Scalability | 10.0 | 6.0 | 5.5 |
| Native Clustering Capabilities | 10.0 | 7.0 | 2.0 |
| Programming Language | 5.0 | 4.0 | 4.5 |
| Core Components | 5.0 | 4.5 | 2.0 |

### J.    Mul

Mul [47–52] is an SDN OpenFlow Controller written in C. It offers a multi-levelled northbound protocol for hosting diverse programmes and supports multi-threading architecture. Among other southbound interfaces, it now supports OpenFlow 1.3, 1.4, and of-config ovsdb. It is designed for dependability and performance, which are crucial in critical networks.

### K.    OpenKilda

OpenKilda [48] is a scalable SDN controller built on web-scale technology from the bottom up. It was designed to deal with insecure control planes that may transport several carriers over long distances. When preparing for a worldwide network, the data plane and

control plane must be scalable. OpenKilda [18] inspired the unstable control plane that may traverse several carriers over long distances. Other SDN controllers have problems with scalability, but this one does not. CloudSmartz is an important strategic and implementation partner for OpenKilda [49]. The company has been concentrating on improving telemetry, network status, self-healing, and the user interface, as well as fixing the control plane scaling issue. In the next installment of this series, we will delve deeper into each of these traits [52,53].

*L.    Maestro*

A Maestro [50–54] can boost throughput by taking advantage of a machine's parallelism. Because Maestro takes care of the difficult issue of managing workloads and thread scheduling, maintaining parallelization requires no effort. It provides APIs for developing modular grid apps that may manage the state of the network and coordinate device interactions.

*M.    Floodlight*

Big Switch Networks' Open Controller is an activity-class, Apache-licensed, Java-based Open-Flow Controller that regulates traffic flow in an SDN environment using the Open-Flow protocol. Floodlights are easy to use, build, maintain, and operate. It may also be used with any hardware or virtual switch that supports the Open Flow Protocol. The Beacon Controller has been forked from the Floodlight [51] and is now open source. Open FloodLight's northbound HTTP API is unencrypted and unauthenticated, allowing a hacker to gain complete control of the network. The Transport Layer Security (TLS) protocol can safeguard data from unauthorized access in SBI, but this is not achievable in SBI. If the Host Tracker service keeps track of all connected hosts with MAC addresses, an intruder can impersonate a legitimate user. As a result, the FloodLight [52] controller is subject to topology interception-based information leaking. Because the host migration is tracked via PACKET-IN messages and no authentication is used, a real user may experience DoS or poor performance. A Floodlight controller addresses this issue since MAC, IP, VLAN ID, and locality are used as indexes in the Host Profile.

*N.    Runos*

Runos [51–53] is a distributed SDN/OpenFlow controller that is open-source. It supports OpenFlow 1.3, and its C++17 implementation ensures performance and scalability. It is a complete user space controller with outstanding features, quick app development, and good performance. For enterprise, datacentre, and professional software-defined networks, use this controller.

*O.    Ryu*

Ryu [40,54] is a Python-based SDN controller that can handle several OpenFlow versions. As well as being able to work in a distributed environment, this type of controller can also be written in the C programming language but suffers from the same performance issues as POX when compared to other types of controllers. It has the benefit of working in a networked environment but does not have the same level of independence as a traditional X-Box controller, and it is an open-source software controller that aims to expand network pliability by simplifying traffic management. Developers can easily design new network management and control applications using Ryu's [55] components, which have comprehensive programme interfaces. Existing components can be updated and merged into existing networks quickly and easily to satisfy the changing needs of different applications that use these components. Because of its fair features, Ryu is a great solution for the smallest commerce and experimentation utilizations. This control layer enables the building of apps and modules because it is developed in Python. However, it is not frequently utilized in real-world applications because of its absence of accessibility and powerlessness to run cross-platform software.

### P. Trema

Trema [56] is a Ruby-based Open-Flow controller framework that provides a variety of network controller features. Trema emphasizes rigorous coding standards to limit the risk of errors and make code maintenance easier. It comes with a network emulator and libraries for building OpenFlow networks on a PC.

### 4.1. Experiment on Mininet

Mininet [56] is a virtual net ambition program for introducing or starting a web that includes switches, routers, end-hosts, connections, and SDN controllers. It is a method of scaling up to hundreds of nodes by utilizing operating system virtualization features and procedures. OpenFlow switches and routers are used. It enables computer-assisted research, development, learning, prototyping, and testing, among other things. After adding a new feature or design, the same coding and testing requirements may be deployed into a major production network by users [57]. Network packets are directed through the SDN controller, where they are processed. Mininet was established by a group of Stanford University scientists to be used as a research and teaching implement. Because of the controller's communication, there is an additional latency on the initial packet [58]. The flows are stored by the switch for a limited time after the first ICMP request and reply. With an Open-Flow controller, a flat Ethernet network with numerous OpenFlow-enabled Ethernet switches, and a large number of hosts linked to those switches, creating a virtual SDN is now straightforward. It comes with functionalities that allow it to work with a variety of controllers and switches [59,60]. Mininet can simulate SDN networks and do testing with the help of a controller. It makes it easy to create complex and customized network topologies to work with multiple concurrent users independently, as shown in Figure 12. It provides Command Line Interface (CLI) support for creating and testing the test beds. It also contains the MiniEdit tool that supports designing and implementing the network through user interface edit has been completed.

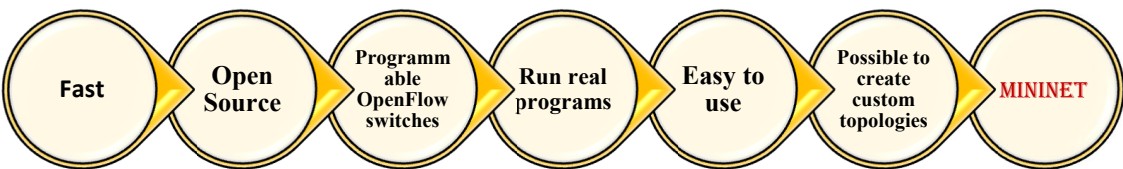

**Figure 12.** Mininet Features.

We worked with Mininet 2.3.0-210211-ubuntu20.04.1-legacy-server-amd64-ovf.zip. running with Intel Core i3-4700MQ CPU processor at a 2.40 GHz × 8, 5.5 GB of Memory. We used VMware (virtualbox.org) to install Ubuntu20.04.1 on a Windows system. Mininet [61–64] is an open-source tool, and the source code can be retrieved using mininet.org URL.

Basics Commands for MININET [12,13]:

○ sudo -s: This command is used for sudo and does not need to be used in every mininet command;

○ sudo mn –h: This command is used to display the mininet help menu;

○ sudo mn: The default topology is the minimal topology, and this command is used for it;

○ sudo mn –c: To clear mininet or a previously used command, use this command;

○ sudo mn –version: This command is used to check mininet's version;

○ mininet> net: This command displays and lists the links in the network that has been formed;

○ Mininet > dump: This command prints the dump information for all nodes in the current mininet network;

○ Mininet > links: This command displays all of the node's link information;

○ Mininet > nodes: This command displays the nodes available for the mininet default minimum topology in the current network;
○ Mininet > pingall: This command will cause each network host to ping every other network host;
○ Mininet > h1 ping h2: The connectivity between hosts h1 and h2 is checked with this command. Until we end the command, it will keep searching for host connectivity;
○ Mininet > h1 ifconfig -a: The IP address, broadcast address, and MAC address of the host h1 will be displayed using this command;
○ Mininet > h1 ip route: This command will display the IP route of the host h1;
○ Mininet > h1 ping -c1 h2: This command tests for a one-packet connection between hosts h1 and h2;
○ Mininet > xterm h2: Connect to the terminal of h2 host, where we can run all the commands;
○ Mininet > exit: This command is used to exit from a created network in mininet.

The default topologies in Mininet are minimal as shown in Figure 13. Single, reversed, linear, and tree, examples are as shown in below Figures 14–18 with command screenshots. We can create a network of all default topology from the given below command in the mininet.

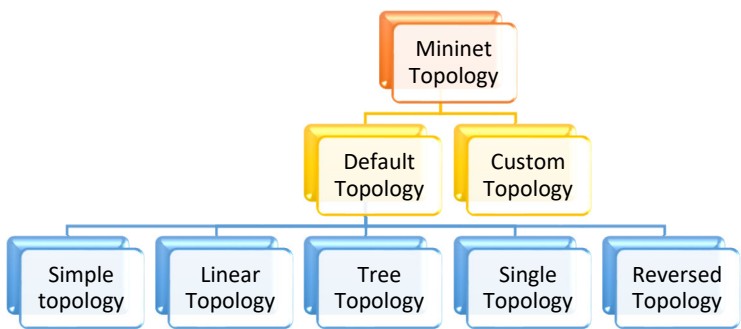

**Figure 13.** Mininet Types.

## Default topologies in Mininet

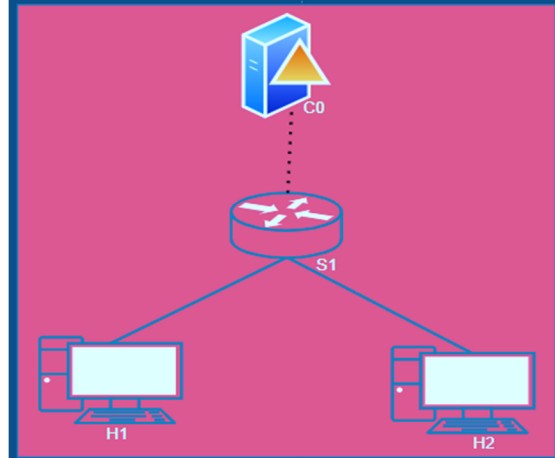

**Figure 14.** Simple topology Mininet command: sudo mn.

```
root@mininet-vm:~# mn --topo single,3
*** Creating network
*** Adding controller
*** Adding hosts:
h1 h2 h3
*** Adding switches:
s1
*** Adding links:
(h1, s1) (h2, s1) (h3, s1)
*** Configuring hosts
h1 h2 h3
*** Starting controller
c0
*** Starting 1 switches
s1 ...
*** Starting CLI:
mininet>
```

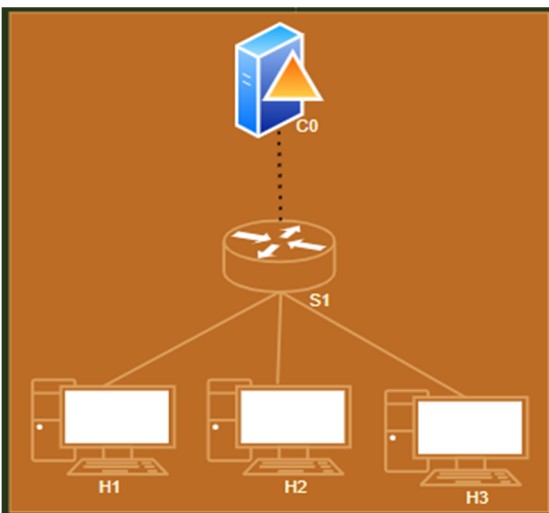

**Figure 15.** Single topology Mininet command: mn –topo single,3.

```
root@mininet-vm:~# mn --topo linear,3
*** Creating network
*** Adding controller
*** Adding hosts:
h1 h2 h3
*** Adding switches:
s1 s2 s3
*** Adding links:
(h1, s1) (h2, s2) (h3, s3) (s2, s1) (s3, s2)
*** Configuring hosts
h1 h2 h3
*** Starting controller
c0
*** Starting 3 switches
s1 s2 s3 ...
*** Starting CLI:
mininet>
```

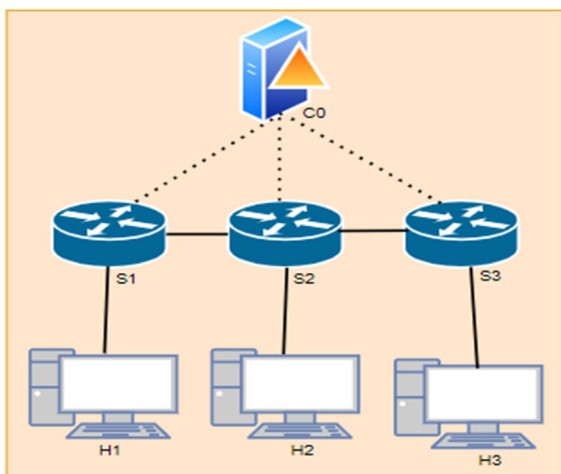

**Figure 16.** Linear topology Mininet command: mn –topo linear,3.

```
root@mininet-vm:~# mn --topo tree,3
*** Creating network
*** Adding controller
*** Adding hosts:
h1 h2 h3 h4 h5 h6 h7 h8
*** Adding switches:
s1 s2 s3 s4 s5 s6 s7
*** Adding links:
(s1, s2) (s1, s5) (s2, s3) (s2, s4) (s3, h1)
 (s5, s7) (s6, h5) (s6, h6) (s7, h7) (s7, h8)
*** Configuring hosts
h1 h2 h3 h4 h5 h6 h7 h8
*** Starting controller
c0
*** Starting 7 switches
s1 s2 s3 s4 s5 s6 s7 ...
*** Starting CLI:
mininet>
```

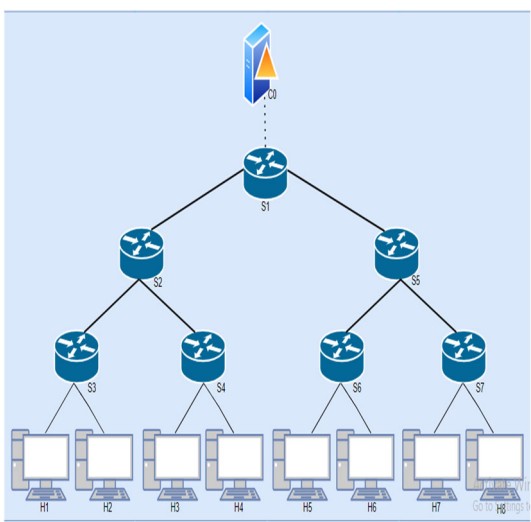

**Figure 17.** Tree topology Mininet command: mn –topo tree,3.

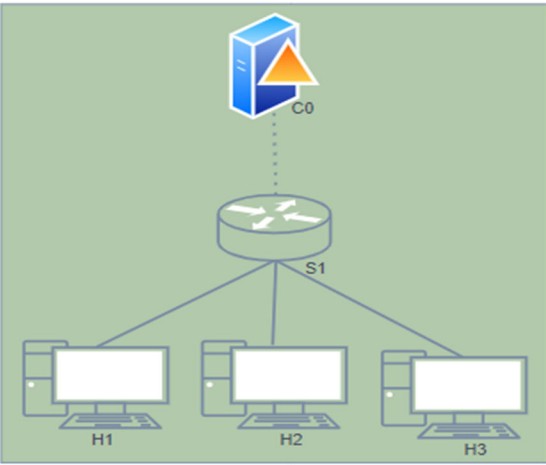

```
root@mininet-vm:/home/mininet# mn --topo reversed,3
*** Creating network
*** Adding controller
*** Adding hosts:
h1 h2 h3
*** Adding switches:
s1
*** Adding links:
(h1, s1) (h2, s1) (h3, s1)
*** Configuring hosts
h1 h2 h3
*** Starting controller
c0
*** Starting 1 switches
s1 ...
*** Starting CLI:
mininet> _
```

**Figure 18.** Reverse topology Mininet command: mn –topo reversed,3.

The new mininet topology class can be defined in the python file for the custom topology. The Topology class must be a subclass of mininet-topo. Topo, which is a built-in class. We can create a network of custom topology from the given below command in the mininet with gedit on the Python platform, as shown in Figures 19 and 20.

```python
"""Custom topology example

Two directly connected switches plus a host for each

  host --- switch --- switch --- host

Adding the 'topos' dict with a key/value pair to generate
topology enables one to pass in '--topo=mytopo' from the ...
"""

from mininet.topo import Topo

class MyTopo( Topo ):
    "Simple topology example."

    def build( self ):
        "Create custom topo."

        # Add hosts and switches
        leftHost = self.addHost( 'h1' )
        rightHost = self.addHost( 'h2' )
        leftSwitch = self.addSwitch( 's3' )
        rightSwitch = self.addSwitch( 's4' )

        # Add links
        self.addLink( leftHost, leftSwitch )
        self.addLink( leftSwitch, rightSwitch )
        self.addLink( rightSwitch, rightHost )

topos = { 'mytopo': ( lambda: MyTopo() ) }
```

```
mininet@192.168.96.130's password:
Welcome to Ubuntu 20.04.1 LTS (GNU/Linux 5.4.0-42-generic x86_64)

 * Documentation:  https://help.ubuntu.com
 * Management:     https://landscape.canonical.com
 * Support:        https://ubuntu.com/advantage

Last login: Sat Apr  2 02:35:53 2022
mininet@mininet-vm:~$  sudo mn --custom ~/mininet/custom/topo-2sw-2host.py --topo mytopo
*** Creating network
*** Adding controller
*** Adding hosts:
h1 h2
*** Adding switches:
s3 s4
*** Adding links:
(h1, s3) (s3, s4) (s4, h2)
*** Configuring hosts
h1 h2
*** Starting controller
c0
*** Starting 2 switches
s3 s4 ...
*** Starting CLI:
mininet> pingall
*** Ping: testing ping reachability
h1 -> h2
h2 -> h1
*** Results: 0% dropped (2/2 received)
```

**Figure 19.** Basic command for custom topology on Mininet.

For create custom simple topology, use this code such as: $ sudo mn –custom ~/mininet/custom/topo-2sw-2host.py –topo mytopo. The methods addHost, addSwitch, addController, addLink function, and others in the class mininet.net.Mininet can be used to design bespoke topologies. The use pattern for the network with custom topology will be as follows:

$ sudo mn -custom <path to the python file .py> –topo topo_name[,arguments..]

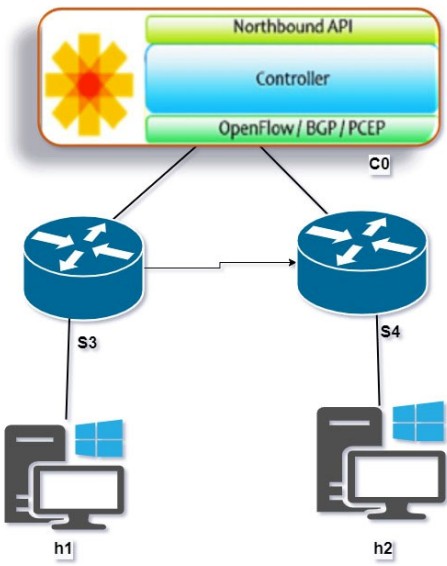

**Figure 20.** Custom simple topology.

### 4.1.1. Custom Topologies (Linear and Tree with ODL Controller)

A tool named Mininet acts as a network emulator and is the best and most straightforward platform for building custom topologies. The host and switch count can be created with the help of this tool. It creates OpenFlow Switches for numerous versions to offer extremely flexible, customized routing in SDN. Version 1.3 of the OpenFlow protocol is used for our experimentation with ODL Controller. The ODL Controller offers a remarkably big platform with a wide range of created modules and services. To build a unique linear and tree custom network topology, execute this code (as shown in Figures 21–24) after installing the ODL Controller. In this experiment, 192.168.174.129 is the IP address of the SDN ODL controller.

```
root@ubuntu:~# sudo mn --controller=remote,ip=192.168.174.129 --switch=ovsk,protocols=OpenFlow13 --mac
 --topo=tree,depth=3,fanout=3
*** Creating network
*** Adding controller
Connecting to remote controller at 192.168.174.129:6653
*** Adding hosts:
h1 h2 h3 h4 h5 h6 h7 h8 h9 h10 h11 h12 h13 h14 h15 h16 h17 h18 h19 h20 h21 h22 h23 h24 h25 h26 h27
*** Adding switches:
s1 s2 s3 s4 s5 s6 s7 s8 s9 s10 s11 s12 s13
*** Adding links:
(s1, s2) (s1, s6) (s1, s10) (s2, s3) (s2, s4) (s2, s5) (s3, h1) (s3, h2) (s3, h3) (s4, h4) (s4, h5) (s
4, h6) (s5, h7) (s5, h8) (s5, h9) (s6, s7) (s6, s8) (s6, s9) (s7, h10) (s7, h11) (s7, h12) (s8, h13) (
s8, h14) (s8, h15) (s9, h16) (s9, h17) (s9, h18) (s10, s11) (s10, s12) (s10, s13) (s11, h19) (s11, h20
) (s11, h21) (s12, h22) (s12, h23) (s12, h24) (s13, h25) (s13, h26) (s13, h27)
*** Configuring hosts
h1 h2 h3 h4 h5 h6 h7 h8 h9 h10 h11 h12 h13 h14 h15 h16 h17 h18 h19 h20 h21 h22 h23 h24 h25 h26 h27
*** Starting controller
c0
*** Starting 13 switches
s1 s2 s3 s4 s5 s6 s7 s8 s9 s10 s11 s12 s13 ...
*** Starting CLI:
```

**Figure 21.** Create custom tree network topology with ODL controller.

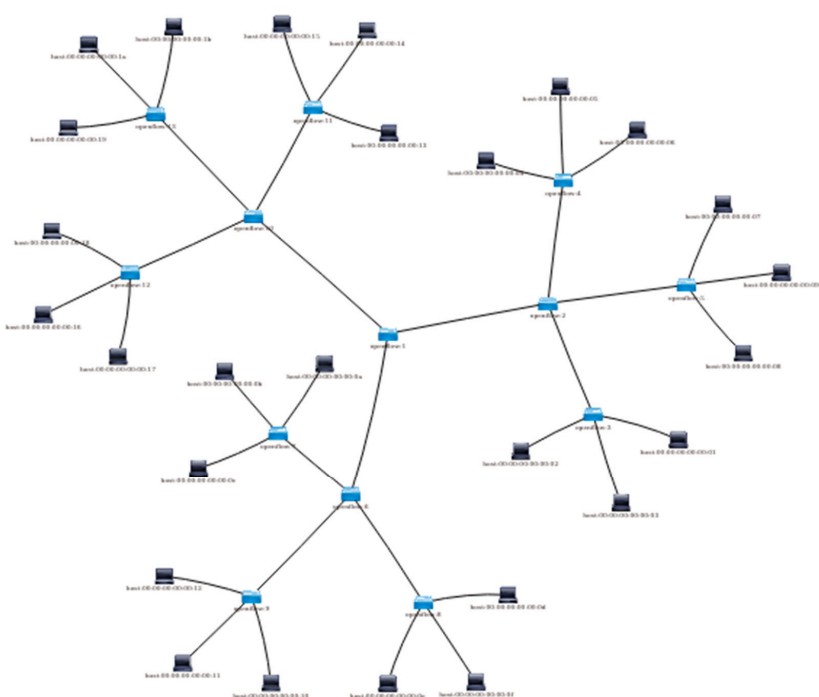

**Figure 22.** Diagrammatically create custom tree network topology on ODL DLUX console.

```
root@ubuntu:~# sudo mn --controller=remote,ip=192.168.174.129 --switch=ovsk,protocols=OpenFlow13 --mac
 --topo=linear,4
*** Creating network
*** Adding controller
Connecting to remote controller at 192.168.174.129:6653
*** Adding hosts:
h1 h2 h3 h4
*** Adding switches:
s1 s2 s3 s4
*** Adding links:
(h1, s1) (h2, s2) (h3, s3) (h4, s4) (s2, s1) (s3, s2) (s4, s3)
*** Configuring hosts
h1 h2 h3 h4
*** Starting controller
c0
*** Starting 4 switches
s1 s2 s3 s4 ...
*** Starting CLI:
mininet> pingall
*** Ping: testing ping reachability
h1 -> h2 h3 h4
h2 -> h1 h3 h4
h3 -> h1 h2 h4
h4 -> h1 h2 h3
*** Results: 0% dropped (12/12 received)
mininet>
```

**Figure 23.** Create custom linear network topology with ODL controller.

Visit your URL as an alternative and use the admin/admin credentials to log into the DLUX interface. Your IP address must be entered in the space given. Following the execution of the code, only switches are visible when you click the upload button on the DLUX console. After running the pingall command, the host will establish a connection, and your network will be ready to process data transfers. An ODL DLUX-based Application that produces and renders a straightforward user interface based on YANG models fed into ODL and aimed to simplify and facilitate application development and testing. Yang visualizer and user interface is a data model language for configuration, state information, network topology (as shown in Figure 25), actions, oxm-container (as shown in Figure 26), and notifications for network elements and services. It is being used to

specify additional data interfaces in Open Daylight and Netconf, an IETF-standard network administration protocol.

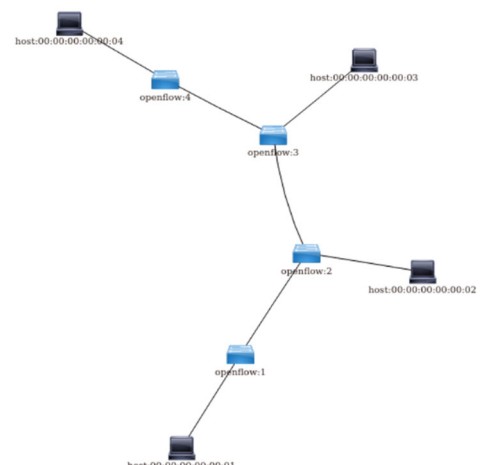

**Figure 24.** Diagrammatically create custom linear network topology on ODL DLUX console.

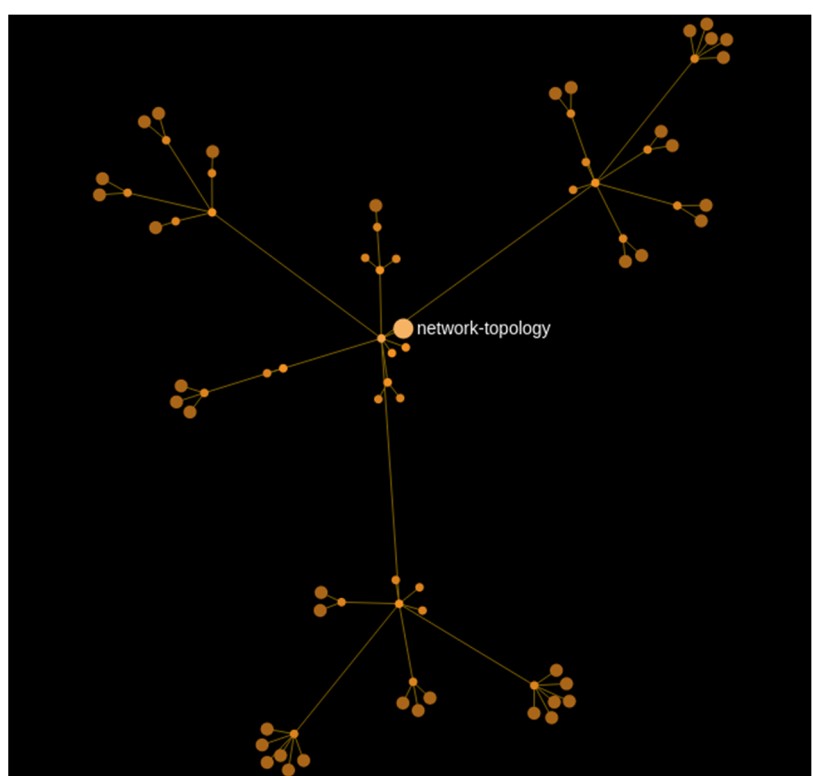

**Figure 25.** Model is network topology on yang-visualizer on DLUX console.

Using the ODL controller to design custom topologies is one of the most advanced engineering techniques in the world. It has been noted that the time it takes to complete the procedure from start to finish is remarkably short—just 775.871 s (13 min 26 pprox..) for tree custom topology and 155.807 s (2.5 min 26 pprox..) for linear custom topology with ODL controller, as shown in Figures 27 and 28.

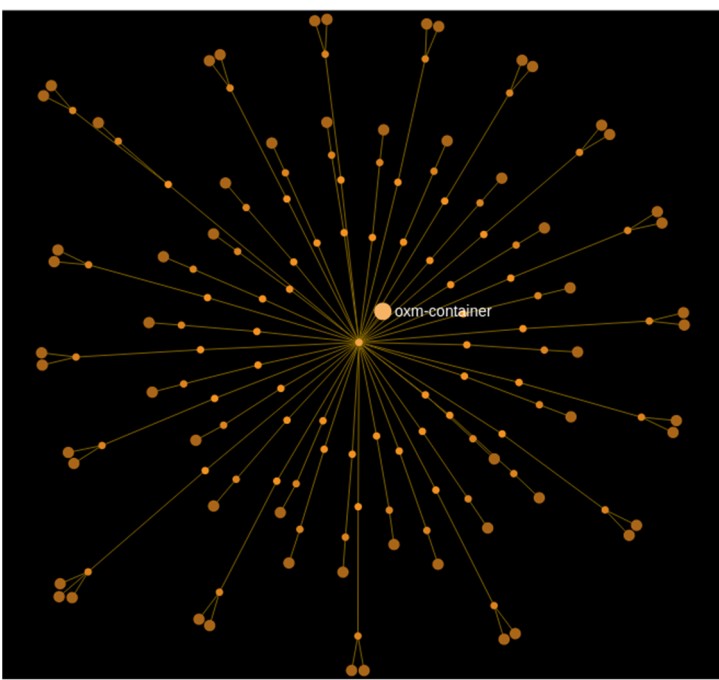

**Figure 26.** Model is oxm-container on yang-visualizer on DLUX console.

```
mininet> exit
*** Stopping 1 controllers
c0
*** Stopping 39 links
.....................................
*** Stopping 13 switches
s1 s2 s3 s4 s5 s6 s7 s8 s9 s10 s11 s12 s13
*** Stopping 27 hosts
h1 h2 h3 h4 h5 h6 h7 h8 h9 h10 h11 h12 h13 h14 h15 h16 h17 h18 h19 h20 h21 h22 h23 h24 h25 h26 h27
*** Done
completed in 775.871 seconds
root@ubuntu:~#
```

**Figure 27.** Tree topology: time to complete the procedure from start to finish with ODL controller.

```
mininet> exit
*** Stopping 1 controllers
c0
*** Stopping 7 links
.......
*** Stopping 4 switches
s1 s2 s3 s4
*** Stopping 4 hosts
h1 h2 h3 h4
*** Done
completed in 155.807 seconds
```

**Figure 28.** Linear topology: time to complete the procedure from start to finish with ODL controller.

### 4.1.2. Custom Topologies (Create Linear and Tree Topology with Ryu Controller)

Version 1.1 of the OpenFlow protocol is used for our experimentation with Ryu Controller, as shown in Figure 29. To build a unique tree custom network topology, after installing the Ryu Controller, follow the below steps to create a custom topology on mininet

with ryu controller. Create tree topology via miniedit (as shown in Figure 30)>Set edit->preference->ok>Controller properties to remote>Click on run>Start ryu-manager on another terminal>Pingall on mininet to check connectivity>exit command run for stop the topology. Create linear topology via miniedit (as shown in Figure 31) and follow the same commands as done for the tree topology.

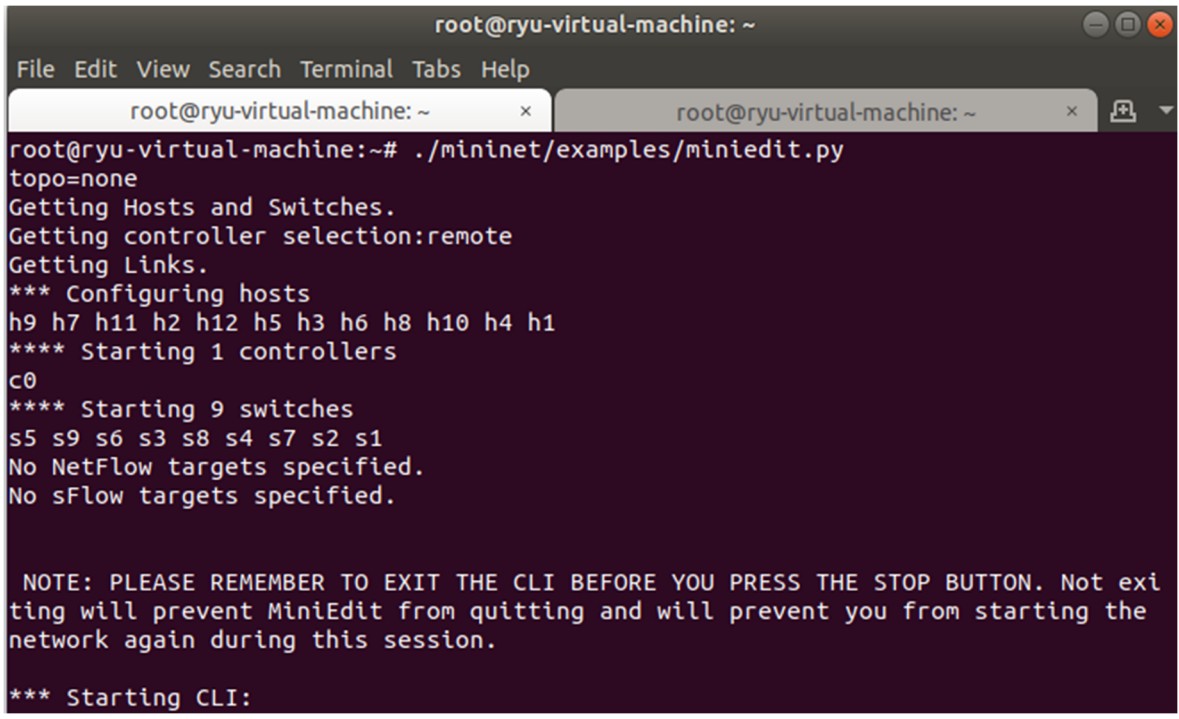

**Figure 29.** Tree custom network topology with Ryu controller.

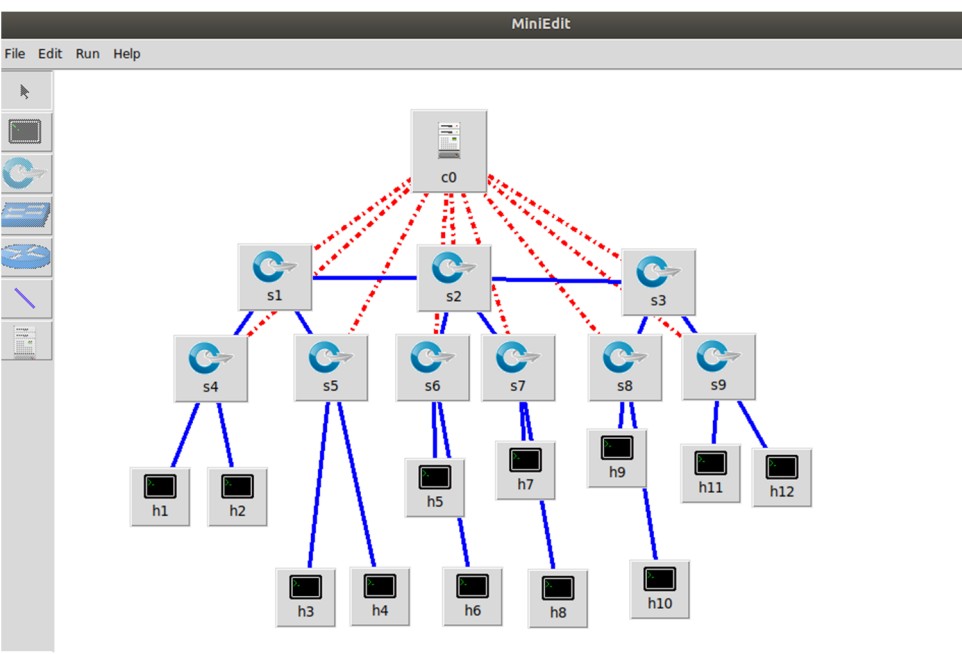

**Figure 30.** Create tree topology via miniedit.

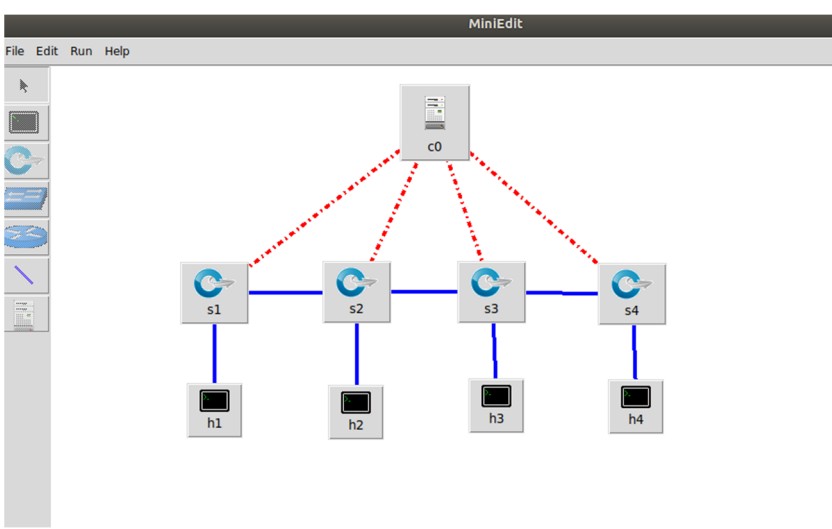

**Figure 31.** Create Linear topology via miniedit.

It has been observed that, when using the Ryu controller, the method can be completed in a very brief amount of time—592.846 s (9.8 min approx.) for tree custom topology and 198.302 s (3.3 min approx.) for linear custom topology.

In this paper, two controllers have been used to build a unique topology. The author concluded that using the ODL controller rather than the Ryu controller makes it simpler to build a customized topology. Using an ODL platform like DLUX Console allows us to gain additional knowledge about network topology. You may also learn how long it takes to start and stop the custom topology from this information, as shown in Table 5. All of these characteristics can be determined with the help of the ODL controller. ODL allows a variety of protocols in its southbound interface because it was designed to reduce vendor locking. ODL has three layers and a number of tasks and features. The following are some significant characteristics of the ODL SDN Controller's layout: The kernel of the ODL, Model-Driven Service Abstraction Layer, employs Yet Another Next Generation (YANG) as the programming model.

**Table 5.** Observation to create custom topologies with ODL and Ryu Controllers.

| Controller | Topology Type | Hosts | Switches | Time (to Complete the Procedure from Start to Finish) |
| --- | --- | --- | --- | --- |
| ODL | Tree | 27 | 13 | 775.871 s (13 min approx.) |
| RYU | Tree | 12 | 9 | 592.846 s (9.8 min approx.) |
| ODL | Linear | 4 | 4 | 155.807 s (2.5 min approx.) |
| RYU | Linear | 4 | 4 | 198.302 s (3.3 min approx.) |

## 5. Features of SDN Controller

The growth of SDN controllers [65–68] is the latest in a series of controllers that seek to discrete control logic from the data surface. This is not the first time that network control has been centralized. Numerous efforts to segregate control logic and data have been attempted since the mid-2000s. SoftRouter [23] and ForCES [24] were introduced in a single network device by Routing Control Platforms (RCP) [22] as an intra-AS (Autonomous System) platform to separate control elements (CEs) from forwarding elements (FEs). The IRSCP (Intelligent Route Service Control Point) [25] provides a dynamic link to enhance network traffic flows. The project's successor, Ethane, provides a more advanced and basic quality management device to the table. In routers, clients can utilize the Path Computation Engine (PCE) [63] to compute paths. Network components are not recognized as a single representation by SANE [27] and Ethane. SDN enables speed and adaptability

by allowing virtualization technology, quick responsiveness to changes in the network, policy implementation, and monitoring of the whole connection from a single location, as demonstrated in Table 6. SDN has grown popular since the release of OpenFlow, the data-plane Application Programming Interface (API) [68–70].

**Table 6.** Use cases and features of controllers.

| Sr. No. | Use Cases | Features | Description |
|---|---|---|---|
| 1 | Legacy Network Interoperability | Efficiency | It is a catch-all word for the various aspects of performance, scalability, dependability, and security. |
| 2 | Network Monitoring | Cross-platform compatibility | Cross-platform compatibility, multithreading, ease of learning, rapid memory access, and effective memory management are all important features of programming languages. |
| 3 | Load Balancing | Interfaces from the Southbound | Southbound APIs provide you with network control. |
| 4 | Traffic Engineering | Northbound Interfaces | The application layer uses the northbound Protocols to communicate with the control system. |
| 5 | Dynamic Network Taps | Partnership | An SDN controller with appropriate partnership oversight has a fair probability of being maintained and improving for a long time. |
| 6 | Multi-layer Network Optimization | Network programmability | The number of linked devices is growing, new services are being launched, and it deals with the unprecedented management challenges of today's networks. |
| 7 | Network Virtualization | Programming Language | A controller can be programmed in a number of languages, including Python, Java, and C++, as well as Ruby, and JavaScript to a degree. |

The most prevalent programming languages for SDN controllers are the following: Python controllers are fast but do not have the same level of memory management as C-coded ones, and they do not support true multithreading. Java controllers are cross-platform and have a lot of modularity, which means they can be written for any platform or operating system. Despite the fact that Python controllers do not support true multi-threading, they have excellent memory management [70–73]. A controller can be programmed in a number of languages, including Python, Java, C++, as well as Ruby and JavaScript to a degree. These languages offer a variety of benefits, including ease of learning, faster memory access, and cross-platform compatibility. When it comes to commercial programmes, Java has a faster runtime.

Network virtualization [65] is one of the most required boons of an SDN. VLANs partition an Ethernet chain into 4094 distinct broadcast domains. Many physical router instances can function at the same time thanks to virtual routing and forwarding (VRF) instances. Server virtualization must abstract and pool network resources in the same way that it abstracts and pools computing resources. Cloud-based apps can now be launched in a production environment without disrupting production traffic—the first time this has happened in the world. These features allow for the establishment of tenant-specific virtual networks that are unaffected by the real network's structure. For security concerns, it is customary for cloud-based service providers to try to isolate one tenant from another. The objective is to ensure that data generated by one group of users do not get into the hands of others. The controller must be able to completely segregate and configure the virtual systems mentioned before. It has to be able to route messages depending on a variety of header information. SDN controllers [66] enable the identification of numerous pathways

from a flow's origin to its destination, as well as traffic splitting across many connections for a single flow. Within a tenant-specific virtual LAN network, the newest version of Cisco's SDN technology allows operators to establish Layer 2 and Layer 3 nets. The constraints of the spanning tree protocol are removed, and the solution's performance and scalability are enhanced. This comprises the capacity to generate Layer 2 networks between endpoints for traffic switching. It also contains the structures below, which authorize an engineer to add features like filtering with intelligence.

OpenFlow is the most well-known southbound API SDN technology, while alternative network device management options include NETCONF, OF-Config, and Opflex. Some routing agreements, such as Intermediate System to Intermediate System (IS-IS), Open Shortest Path First (OSAPIPF), and Border Gateway Protocol, are being developed as southbound interfaces (BGP).

The northbound interfaces are the most important part of the SDN controller infrastructure. Because of their relevance, northbound APIs must serve a broad range of applications. They are also compatible with cloud management platforms such as OpenStack and Cloud Stack. The Representational State Transfer (REST) protocol appears to be the most often utilized northbound interface at the present time.

The first established southbound interface was the OpenFlow protocol [67]. Critical elements like as IPv6 support are not included in the OpenFlow v1.0 standard. It allows you to manage the forwarding plane of OpenFlow switches directly. Before acquiring one, we must study the controller's OpenFlow capability as well as the roadmap for future Open Flow adaption. SDN solves the enormous management issues that today's networks face. Automated scripts may be executed using command-line interfaces (CLIs), and apps can be installed above the controller platform. According to IT professionals, traditional static network device maintenance is wasteful, error-prone, and results in inconsistency.

The term efficiency refers to the various performance, scalability, dependability, and security parameters. A controller's performance [68] is determined by a variation of elements, as well as the number of interfaces it can handle. The majority of the time and effort spent comparing controllers is spent measuring performance. The distributed SDN strategy, which is supported by a variety of controllers, tries to address the issue of control centralization in the SDN system. Scalability, dependability, and security are also featuring of an SD-WAN network. If an SDN controller is closely monitored by a good partner, it has a better chance of being supported and expanded in the long run. Experience in the network and computer industries, as well as the partner's firm's financial competence, are the most important factors in determining trust and product use. Cisco, the Linux Foundation, Intel, IBM, Juniper, and other well-known companies have all made SDN investments. When comparing controllers, the bulk of the above characteristics is considered.

Typical LANs are not scalable when servicing east and west traffic. Layer 3 routing capabilities are used to join numerous Layer 2 networks [69] in a typical LAN's multi-tiered design. Typical LANs struggle to manage this sort of traffic since at least one Layer 3 device, and most likely several Layer 3 devices, are in the end-to-end path. SDNs may be used by IT organizations to handle all networking functions as if they were one device. When it comes to SDN scalability, the number of switches that an SDC controller can support is essential. In the present context, IT organizations should expect their controllers to service at least a hundred switches. The ability to build a network that spans several places is required of an SDN controller. The impacts of online broadcast must be prevented by network SDN [70] controllers. Another factor that limits an SDN's scalability is the flow table entries proliferate. A controller might leverage Reworks of protocols in the program's base to do this. This capability allows you to relocate virtual machines and virtual storage between locations. The controller must additionally apply the network policies for routing and forwarding to the relocated servers and/or storage.

An SDN controller can handle an infinite number of flows per second. Figure 32 shows the ten key factors of controller functionality of Software Defined Networking. More controllers must be installed if a production SDN switch in a production network

initiates more flows than its current controller(s) can handle. There are two types of flows: proactive and reactive. Performance measurements are critical when extra SDN controllers are required. A packet arrives at an OpenFlow [71–73] switch that does not match any of the entries in its flow table. A reactive flow setup occurs when packets are sent in the opposite direction rather than the other way around. The controller knows what to do with the packet since it is programmed before it reaches the Open Flow control. SDN controllers should not slow down the generation and deletion of flow entries handled by the switch. The processing capability of the switches connected to the controller, as well as the controller's processing and I/O capabilities, determine the flow setup time.

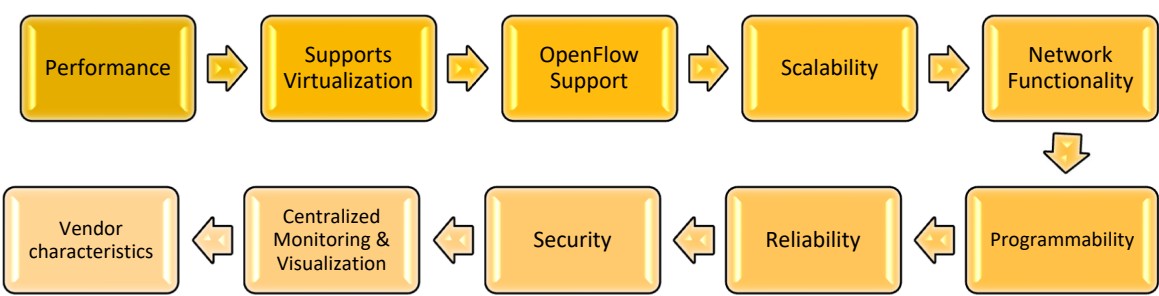

**Figure 32.** SDN controller functionality: 10 key factors.

Traditional network programmability methods are inefficient, time-consuming, and prone to errors. The compositions do not vary as the system conditions change; hence, this strategy is excessively stable. A programmable SDN controller is required to ensure that all devices are connected at the same time. This means the controller has three programmable [72] interfaces. IT departments should look into traffic rerouting options. For security reasons, an organization could choose to bypass a firewall for inbound traffic to a server. SDN controllers should be able to apply advanced packet filters, which are akin to smart, dynamic ACLs. By providing templates for constructing CLIs for network dynamic programming, an SDN controller can aid programmability.

SDN controllers centralize control information and make it available to an endless number of SDN applications. SDN controllers should be able to apply advanced packet filters. These filters are similar to smart, dynamic ACLs, and they serve as a starting point for creating CLIs for network dynamic programming [69]. Traditional network functions like gateways and application servers, as well as orchestrating systems like OpenStack, might be included in these apps.

The use of an SDN controller can aid in the improvement of network reliability. In the incident of a connection failure [61], it must be able to immediately redirect traffic to a graphic link. This demands the monitoring of network topology [70–73] across multiple networks. The controller's functions must be understood by IT organizations. For future generations of routers and other devices, the controller [14] must provide technology and design alternatives to standard wired and wireless connectivity. Both hardware and software redundancy are required by an SDN controller. The controller must support clustering in order to provide fast failover times. The active/hot-standby mode of two SDN controllers improves reliability, scalability, and performance.

An SDN controller must offer security and verification for businesses of network governors (APDIN), according to the Association for Data Protection in Networks. SDN control traffic can be restricted using the same mechanisms as other essential types of traffic, such as management traffic, by the controller. An SDN controller must be able to apply intelligent packet filters, send rate-limit control messages, and notify network managers if the network is under assault. According to the controller, all tenant-sharing infrastructure must be completely isolated from all other tenants [17].

According to centralized monitoring [19] and controller visualization, when an application's performance degrades, the end user observes it first, not the IT staff. One of the

most prevalent reasons why IT organizations are unaware of poor application performance is a lack of understanding of end-to-end network flows. sFlow's scalability is achieved by sampling. This indicates that an aggregate of 1 out of N data packets is periodically examined depending on a pre-set sampling rate. It lacks the level of information required by IT organizations to comply with a wide range of government and industry standards. An SDN provides IT organizations with end-to-end network flow visibility. Unlike sFlow, OpenFlow can collect counters on any defined flow. An SDN [73] control plane should be clever to detect network failures and automatically change the direction of a flow.

An SDN controller [29] must be able to detect and see the arrangement's physical connections to an IT organization, as well as acquire more detailed information on each flow. Furthermore, the controller should enable IT organizations to view flows from both a physical and virtual network perspective. The SDN controller is a critical piece of hardware that allows IT managers and business users to access a wide range of network data. The controller's ability to be monitored using industry-standard protocols and methodologies should be a priority for IT teams. SNMP should be able to monitor the status of devices linked to the controller, for example [30–33].

The SDN sector, in general, and the market for SDN controllers in particular, are quickly expanding. IT organizations should consider the vendor's overall capabilities as well as the technical team's depth. It is crucial to think about not just the controller's technical features, but also the supplier's suppliers and customer service and support options [73]. A simple litmus test that IT organizations should use is the following: Is the vendor financially and technically capable of maintaining the ongoing development that will be linked with SDN? The vendor's resilience in the SDN market is an important consideration that companies should consider. Choosing a larger company's SDN controller [40,41] provider may be less dangerous, but the IT department and the organization's reputation are still at risk.

## 6. Conclusions

Software Defined Networking (SDN) [19] controllers come in a variety of flavors, and some are more suited to specific scenarios than others. There is no universally acknowledged answer, and there is unlikely to be in the future. This is because researchers and the networking industry are still defining use cases for SDN controllers [65]. The first thing to consider when selecting a controller is its intended use. Opendaylight, for instance, makes it much simpler to move and learn to design apps than other controllers' selection of the greatest fit controller suits an app-dependent procedure required for the range of applications and controllers. As a result, this study compares the effects of various SDN controllers on SDN.

In this paper, we compared a number of different controllers. OpenDaylight has shown to be the best choice as a full-detailed SDN controller, as shown in Table 7. A custom topology on a mininet was created using the ODL platform [66]. The accessibility of numerous SDN controllers generates a deal of ambivalence when it comes to choosing the right controller. It has the potential to become a useful controller because it braces a wide range of applications and has a strong environment. We used VMware (virtualbox.org) to install Ubuntu20.04.1 on a Windows system. The ODL controller [73] has been shown to be superior to the SDN controller in terms of latency, detain, and uneasiness for a range of real-world scenarios, as well as when simulated in mininet. Other topologies with more composite structures can be modelled using mininet to gain further insight into their impact on SDN performance. This research effort will be useful for various network managers to analyze the performance of their networks, as these controllers can be controlled by a variety of software and hardware. In addition, the controllers can execute a variety of programmes to make the outcomes more dynamic and lifelike. When the aforementioned controllers are used in a real network, they will provide an excellent basis for analyzing performance on a sizeable scale. The ODL is a state-of-the-art controller for delivering security features on SDN networks, allowing future firewall applications and new research

areas to be built. Future comparisons of SDN controllers could be enhanced by putting into consideration additional implementations, such as ODL or RYU (this controller even has a built-in network emulator, but simulations using mininet would also be necessary for the tests to be accurate and applicable). Numerous topologies with complex scenarios can be simulated in mininet to broaden the applicability of the results. The controllers can also execute additional programmes to improve the outcomes' accuracy and dynamic nature.

**Table 7.** Controllers with TLS, PD, MT, and RAPI.

| Controllers | ODL | ONOS | NOX | POX | RYU | Beacon | Floodlight | Trema | MUL |
|---|---|---|---|---|---|---|---|---|---|
| Year | 2014 | 2014 | 2009 | 2013 | 2013 | 2010 | 2013 | 2011 | 2012 |
| Language | Java/Python | Java | C++ | Python | Python | Java | Java | Ruby/C | C/Python |
| Transport Layer Security | ✓ | ✗ | ✗ | ✗ | ✓ | ✗ | ✓ | ✗ | ✓ |
| Physically Distributed | ✓ | ✓ | ✗ | ✗ | ✗ | ✗ | ✗ | ✗ | ✓ |
| Multi-Threaded | ✓ | ✓ | ✗ | ✗ | ✗ | ✓ | ✓ | ✗ | ✓ |
| Rest API | ✓ | ✗ | ✗ | ✗ | ✗ | ✗ | ✓ | ✗ | ✓ |

**Author Contributions:** Conceptualization: N.G., M.S.M., S.T. and S.B. (Salil Bharany); methodology: N.G., M.S.M., S.T., S.B. (Sumit Badotra) and S.B. (Salil Bharany); validation: S.B. (Sumit Badotra) and S.B. (Salil Bharany); formal analysis: S.B. (Sumit Badotra), S.B. (Salil Bharany) and M.A.; investigation: S.B. (Salil Bharany), S.B. (Sumit Badotra) and N.G.; resources: M.A., M.S.M. and S.B. (Salil Bharany); data curation: N.G., M.S.M., S.T., S.B. (Salil Bharany), M.A. and S.B. (Sumit Badotra); writing original draft preparation: N.G., M.S.M., S.T. and S.B. (Salil Bharany); writing review and editing: N.G., M.S.M., S.T., S.B. (Salil Bharany) and S.B. (Sumit Badotra); visualization: S.B. (Salil Bharany); supervision: M.S.M., S.T., S.B. (Sumit Badotra) and M.A.; project administration: M.S.M., S.B. (Salil Bharany) and M.A.; funding acquisition: M.S.M. and M.A. All authors have read and agreed to the published version of the manuscript.

**Funding:** This research is funded by King Saud University, RSP2022R459.

**Acknowledgments:** We deeply acknowledge King Saud University for supporting this study Research Supporting Project number (RSP2022R459), King Saud University, Riyadh, Saudi Arabia.

**Conflicts of Interest:** The authors declare no conflict of interest.

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
