# Peer review of "A Comparative Study of Software Defined Networking Controllers Using Mininet"

_electronics, doi:10.3390/electronics11172715_

Round 1

Reviewer 1 Report

In this research, paper authors run an experiment to show how to use ODL to establish a custom network topology on a Mininet. The experimental results show that the O controller, with its larger bandwidth and reduced latency, outperforms other controllers in all topologies (both the default topology and a custom topology with ODL). The research flow of the paper is novel and quite interesting. The overall contribution is also found satisfactory. The following are my key observations/suggestions.

1.       Try to summarise existing work of ODL comparison in a table

2.       There are a lot of typos and grammatical mistakes, rectify them.

3.       What was the basis of segregation for the authors and their cumulative numbers of research articles related to SDN controllers??

4.       Figures quality is very low, improve them.

5.       Add a paragraph of major findings from section 3.

6.       In Section no. 5 have you included all features of the SDN Controller??

7.       OpenDaylight [44-46] has been shown to make the best choice as a full-detailed SDN controller, as shown in Table 6….Where is Table 6????

8.       Check for references missing information and relevant format.

Author Response

Reviewer-1 Queries

Authors Response

Try to summarise existing work of ODL comparison in a table

In the updated manuscript we Done (on page number 15), at line 395 to 398

There are a lot of typos and grammatical mistakes, rectify them.

In the updated manuscript It has been corrected

What was the basis of segregation for the authors and their cumulative numbers of research articles related to SDN controllers??

In this paper the basis of segregation for the authors is given below:

1.     SDN architecture & its work flow

2.     Related work on SDN (search is restricted from 2012 to 2022)

3.     Comparison & features of SDN Controllers

4.     Different topologies created with ODL and Ryu controller

Figures quality is very low, improve them.

Resolution has been Increased

Add a paragraph of major findings from section 3.

In the updated manuscript a new Paragraph has been added

In Section no. 5 have you included all features of the SDN Controller??

Yes, we have included all the features

OpenDaylight [44-46] has been shown to make the best choice as a full-detailed SDN controller, as shown in Table 6…. Where is Table 6????

In line 467, below table 5

Check for references missing information and relevant format.

In the updated manuscript we have Checked and added all relevant references.

Reviewer 2 Report

This paper conducts a comparative study on well-known SDN controllers, e.g., Ryu, ODL, Floodlight, and ONOS, using Mininet and shows how they differ from one another via Mininet-based experiments. The authors claim the ODL controller achieves relatively better performance than the others in terms of performance indicators such as bandwidth and latency. The topic is interesting and the paper is easy to follow. Some suggestions are listed below to improve the quality of the paper:

1.  The reference list is a bit out-of-date and some key references seems missing. Please cite more high-quality works regarding SDN and its applications, e.g., https://doi.org/10.1016/j.comnet.2022.109098 and https://doi.org/10.1016/j.jnca.2016.04.005. 

2. Some figures are of low resolution, e.g. Fig. 1, Fig. 2, Fig. 3, Fig. 4, and Fig. 8. Please replace them with higher resolution ones.

3. Those figures reporting the original experimental results are not necessary, including Figs. 13, 15, 17, 19, 21, 23, 25, 27, 31 and 32. Instead, please provide statistical results (tables and figures) based on the original ones since statistics are more concise and direct.

4. There are some grammar and typo errors throughout the paper. Please correct them.

Author Response

Reviewer-2 Queries

Authors Response

This paper conducts a comparative study on well-known SDN controllers, e.g., Ryu, ODL, Floodlight, and ONOS, using Mininet and shows how they differ from one another via Mininet-based experiments. The authors claim the ODL controller achieves relatively better performance than the others in terms of performance indicators such as bandwidth and latency. The topic is interesting and the paper is easy to follow. Some suggestions are listed below to improve the quality of the paper:

1.  The reference list is a bit out-of-date and some key references seems missing. Please cite more high-quality works regarding SDN and its applications, e.g., https://doi.org/10.1016/j.comnet.2022.109098 and https://doi.org/10.1016/j.jnca.2016.04.005. 

In the updated manuscript all the concerns are Checked & Updated

2. Some figures are of low resolution, e.g. Fig. 1, Fig. 2, Fig. 3, Fig. 4, and Fig. 8. Please replace them with higher resolution ones.

In the updated manuscript images are replaced with higher resolution

3. Those figures reporting the original experimental results are not necessary, including Figs. 13, 15, 17, 19, 21, 23, 25, 27, 31 and 32. Instead, please provide statistical results (tables and figures) based on the original ones since statistics are more concise and direct.

These images are included with screenshot code because a detailed explanation of how to design default and custom topologies is included in this paper.

4. There are some grammar and typo errors throughout the paper. Please correct them.

In the updated manuscript ,It has been corrected

Reviewer 3 Report

Figure 1: The color of the texts is not well distinguishable -- The red texts and the orange background; the orange texts and the white background; and the gray texts with the light yellow backgrounds. 

L83: an apps that be -> an app that is 

Figure 3, 7: font's colors. 

Line 253: what are the keywords and indices that are used for the search? I see some terms in Fig.34(a)-34(c). Are they all the keywords used by this paper? Please explicitly indicate this. 

More importantly, for Fig9(a)-9(f), what is the purpose of showing those figures? 

What conclusions/recommendations can be given based on those statistics from a single database, the Scopus electronic database? 

Please explicitly explain and provide the insight behind those figures. 

In Section 4: 

Again, the conclusion/recommendations of the mininet experiments are missing. In other words, what are the intriguing findings via your mininet experiments? 

Please provide your insights and a good takeaway from the mininet experiment. 

In Section 5: 

The "features of controllers" listed in Table 3 are most well-known to the community already. I understand that the authors compile the information here for better reference. 

I would appreciate it if the authors can add sub-titles to each feature discussion to make the large numbers of paragraphs (7 pages) more friendly to read. 

In table 3: I feel that organizing the long section 5 in these 7 might help improve the presentation of the topic "features of controllers". 

Section 6: 

I feel the following statement about the ODL lacks evidence from the paper. 

"The ODL controller has been shown to be superior to the SDN controller in 780 terms of latency, detain, and uneasiness for a range of real-world scenarios, as well as simulated 781 in mininet."

Basically, I see that mininet experiments are run using the ODL but I can't find any comprehensive comparison with the experiments about the latency, detain, and uneasiness among all other SDN controllers with the ODL. 

Please point those pieces of evidence out in the response to my review if I have missed them. 

Author Response

Reviewer-3 Queries

Authors Response

Figure 1: The color of the texts is not well distinguishable -- The red texts and the orange background; the orange texts and the white background; and the gray texts with the light-yellow backgrounds. 

In the updated manuscript it has been Implemented as per suggestions

L83: an apps that be -> an app that is

In the updated manuscript it has been Replaced

Figure 3, 7: font's colors.

In the updated manuscript the  concern have been corrected .

Line 253: what are the keywords and indices that are used for the search? I see some terms in Fig.34(a)-34(c). Are they all the keywords used by this paper? Please explicitly indicate this.

In the updated manuscript ,Added as recommended

More importantly, for Fig9(a)-9(f), what is the purpose of showing those figures?

The work done on comparison of SDN controllers in the last 9 years

What conclusions/recommendations can be given based on those statistics from a single database, the Scopus electronic database?

Please explicitly explain and provide the insight behind those figures.

Explained (on page number 9), at line number 263 to 275

In Section 4: 

Again, the conclusion/recommendations of the mininet experiments are missing. In other words, what are the intriguing findings via your mininet experiments? 

Please provide your insights and a good takeaway from the mininet experiment.

Two controllers have been used to build a custom topology (ODL & Ryu). The author concluded that using the ODL controller rather than the Ryu controller makes it simpler to build a customised topology. Using an ODL platform like DLUX Console allows us to gain additional knowledge about network topology. You may also learn how long it takes to start and stop the custom topology from this information. All of these characteristics can be determined with the help of the ODL controller.

In Section 5: 

The "features of controllers" listed in Table 3 are most well-known to the community already. I understand that the authors compile the information here for better reference. 

I would appreciate it if the authors can add sub-titles to each feature discussion to make the large numbers of paragraphs (7 pages) more-friendly to read. 

In the updated manuscript I has been Added – line number 673 (There is no need for subtitles because every feature is described in each paragraph)

In table 3: I feel that organizing the long section 5 in these 7 might help improve the presentation of the topic "features of controllers". 

Features & Functionality of SDN controllers as stated in section 5.

Section 6: 

I feel the following statement about the ODL lacks evidence from the paper. 

"The ODL controller has been shown to be superior to the SDN controller in 780 terms of latency, detain, and uneasiness for a range of real-world scenarios, as well as simulated 781 in mininet."

Basically, I see that mininet experiments are run using the ODL but I can't find any comprehensive comparison with the experiments about the latency, detain, and uneasiness among all other SDN controllers with the ODL. 

Please point those pieces of evidence out in the response to my review if I have missed them.

Using the ODL-controller to design custom topologies is one of the most advanced engineering techniques in the world. It has been noted that the time it takes to complete the procedure from start to finish is remarkably short—just 775.871 seconds (2.5 minutes approx.) for tree custom topology and 155.807 seconds (13 minutes approx.) for linear custom topology with ODL controller, as shown in figure 31 and figure 32.

Using the ODL controller rather than the Ryu controller makes it simpler to build a customised topology. Using an ODL platform like DLUX Console allows us to gain additional knowledge about network topology. You may also learn how long it takes to start and stop the custom topology from this information. All of these characteristics can be determined with the help of the ODL controller.

Round 2

Reviewer 2 Report

The revised manuscript answered the comments. It can be considered for publication now.

Author Response

Reviewer-1 Queries

Authors Response

The revised manuscript answered the comments. It can be considered for publication now

I am very grateful to receive this comment.

Reviewer 3 Report

I appreciate the explanations provided by the authors for the statistics of the SDN literature. 

In section 4, I see that the authors added an experiment using the Ryu SDN controller. The extra experiment and the comparison make more sense to me to draw the conclusion,  "the author concluded that using the ODL controller rather than the Ryu controller makes it simpler to build a customized topology". 

In the authors' response to my Section 6's comment. 

Figure 31 and Figure 32 in the updated manuscript don't give the performance number (775.871 and 155.807 seconds.) They should be Fig. 26 and Fig. 27 in the updated manuscript. 

But I did not find comparable numbers in the Ryu case. 

I would appreciate the authors can obtain those numbers in the Ryu setup, such that the claim of the ODL controller being simpler is more convincing. 

In summary, the authors addressed most of my review comments and concerns except that I would expect a completion time for the Ryu setup compared to the ODL ones (775.871 and 155.807 secs). 

Author Response

Reviewer-2 Queries

Authors Response

I appreciate the explanations provided by the authors for the statistics of the SDN literature. 

In section 4, I see that the authors added an experiment using the Ryu SDN controller. The extra experiment and the comparison make more sense to me to draw the conclusion, “the author concluded that using the ODL controller rather than the Ryu controller makes it simpler to build a customized topology". 

In the authors' response to my Section 6's comment. 

Figure 31 and Figure 32 in the updated manuscript don't give the performance number (775.871 and 155.807 seconds.) They should be Fig. 26 and Fig. 27 in the updated manuscript. 

But I did not find comparable numbers in the Ryu case. 

I would appreciate the authors can obtain those numbers in the Ryu setup, such that the claim of the ODL controller being simpler is more convincing. 

In summary, the authors addressed most of my review comments and concerns except that I would expect a completion time for the Ryu setup compared to the ODL ones (775.871 and 155.807 secs).

Using the ODL controller:  It has been observed that the time it takes to complete the procedure from start to finish is remarkably short—just 775.871 seconds (13 minutes approx.) for tree custom topology and 155.807 seconds (2.5 minutes approx.) for linear custom topology (In line number 587 to 591).

Using the RYU controller: It has been observed that, the method can be completed in a very brief amount of time—592.846 seconds (9.8 minutes approx.) for tree custom topology and 198.302 seconds (3.3 minutes approx.) for linear custom topology. (In line number 610 to 612).

Observation: Both controllers have been used to generate custom tree and linear topologies, although the ODL controller produced a tree topology with more hosts and switches than the Ryu controller. However, in both ODL and RYU, linear topology contains equal hosts and switches.

However, it appears that the Ryu controller-created tree topology requires more time to finish the entire procedure.

Controller

Topology type

Hosts

Switches

Time (to complete the procedure from start to finish)

ODL

Tree

27

13

775.871 seconds (13 minutes approx.)

RYU

Tree

12

9

592.846 seconds (9.8 minutes approx.)

ODL

Linear

4

4

155.807 seconds (2.5 minutes approx.)

RYU

Linear

4

4

198.302 seconds (3.3 minutes approx.)